# Efficiently Scanning and Resampling Spatio-Temporal Tasks with Irregular Observations

## Abstract

Various works have aimed at combining the inference efficiency of recurrent models and training parallelism of MHA for sequence modeling. However, most of these works focus on tasks with fixed-dimension observation spaces, such as individual tokens in language modeling or pixels in image completion. Variably sized, irregular observation spaces are relatively under-represented, yet they occur frequently in multi-agent domains such as autonomous driving and human-robot interaction. To handle an observation space of varying size, we propose a novel algorithm that alternates between cross-attention between a 2D latent state and observation, and a discounted cumulative sum over the sequence dimension to efficiently accumulate historical information. We find this resampling cycle is critical for performance. To evaluate efficient sequence modeling in this domain, we introduce two multi-agent intention tasks: simulated agents chasing bouncing particles and micromanagement analysis in professional StarCraft II games. Our algorithm achieves comparable accuracy with a lower parameter count, faster training and inference compared to existing methods.

## 1 Introduction

Spatio-temporal modeling tasks with complex unstructured or semi-structured state and observation spaces can be identified in a variety of domains. Designing deep learning algorithms to excel at these tasks requires deliberate handling of both the accumulation of knowledge from historical observations, and the summarization of current observations, which can be computationally expensive. This is a particular concern in multi-agent domains such as motion prediction and behavior modeling, which have real-time requirements and often need to be performed using relatively low compute resources available on an edge devices. Recurrency is a popular paradigm in deep learning as it naturally maps onto problems which are sequential and causal in nature (Werbos, 1990). While iterative processing of data with recursion can be efficiently performed in $\mathcal{O}(1)$ with respect to sequence length $L$, transformer (Vaswani et al., 2017) and convolution (CNN) methods introduce compute and memory complexity $\mathcal{O}(L^2)$ and $\mathcal{O}(L)$ respectively. However, transformer and CNN algorithms are parallelizable at training time, efficiently utilizing hardware and invoking backward propagation paths uncorrelated with sequence length. This leads to a trade-off between compute efficiency at training or inference time.

Proposals to address this trade-off can be grouped into various categories that aim to take advantage of inference efficiency of recurrence and parallelization at training time. One avenue is to address the $\mathcal{O}(L^2)$ complexity of multi-head attention (MHA) by introducing variations with linearized, amortized attention or windowed attention with recurrence (Sun et al., 2023; Peng et al., 2023; Katharopoulos et al., 2020; Hutchins et al., 2022; Zhai et al., 2021; Didolkar et al., 2022). State space models (SSMs) (Dao & Gu, 2024; Gu et al., 2022; Fu et al., 2023; Gu et al., 2020) are presented as a compelling alternative to transformers, demonstrating efficacy in a variety of long-range dependency tasks and language modeling. SSMs can be formulated as a convolution for training parallelism or as a recurrent model for inference over long sequences. In each case, these methods are evaluated on sequences with a fixed dimensional observation space $\mathcal{O} \in \mathbb{R}^d$, such as tokenized text, image pixels, audio spectrogram data and time-series data (Tay et al., 2021; Wu et al., 2021). Focusing on these sequences leaves a blind-spot in tasks with an irregular observation spaces $\mathcal{O} \in \mathbb{R}^{d(t)}$ such those in as multi-agent interactions, where the number of agents and links between these may vary over time (Ettinger et al., 2021; Vinyals et al., 2017).

In this work, we investigate a range of encoding and state space modeling approaches to handle irregular observation settings to provide clarity around the most efficient and best performing approaches. We propose a novel and efficient algorithm that utilizes a 2D latent state and alternates between input sampling, and accumulating historical information as a weighted sum (inclusive scan). This weighted sum can be performed efficiently in parallel on a GPU with an inclusive-scan Merrill & Garland (2016) during training, and incrementally during inference. We show that the resampling cycle is more effective than a continued self-attention block, or not alternating between accumulation and processing. This method natively supports a two dimensional latent state. We benchmark against a range of baselines including transformers (Vaswani et al., 2017; Sun et al., 2023), recurrent neural networks (RNNs) (Chung et al., 2014; Hochreiter, 1997; Martin & Cundy, 2018), and a State-Space Model (SSM) (Dao & Gu, 2024).

To test the efficacy of these training and inference efficient algorithms on tasks with more complex observation spaces, we use two multi-agent interaction benchmarks. The first is a "gymnasium" style simulation that involves agents chasing randomly assigned particles[1]. The second benchmark is based on *StarCraft II* (SC2), a real-time strategy video game, where we extract instances of players in combat. Each of these tasks involves a multidimensional time-varying observation space. In summary our contributions are as follows:

- The introduction of two multi-agent interaction challenges to better evaluate sequence modeling algorithms with irregular observation spaces.
- A novel algorithm to efficiently address sequence modeling tasks with irregular observation spaces. We find that this algorithm achieves comparable accuracy to alternatives with a lower parameter count and improved throughput in training and inference regimes.
- Empirically comparing a combination of two encoders and three algorithms to reduce an irregular observation space to a fixed-size amenable for sequence modeling.

## 2 BACKGROUND

**Efficient and parallelizable sequence modeling** is of great interest to the research community, as attaining efficient utilization of parallel computation at training time while performing inference efficiently are desirable attributes of sequence modeling algorithms. Previously, a trade-off had to be considered between using a model that is more efficient to train, or run inference. Practitioners could choose between $\mathcal{O}(1)$ inference efficiency with an RNN, or training parallelism with a CNN or Transformer. With the introduction of SSMs that can be formulated as either a convolution or recurrency, practitioners can take advantage of both efficient incremental inference, and training time parallelism, while also avoiding the $\mathcal{O}(L^2)$ complexity of transformers. While transformers have demonstrated effective adaptability to variety of domains of varying dimensionality and sparsity (Vaswani et al., 2017; Dosovitskiy et al., 2021; Zhu et al., 2022; Yuan et al., 2021), efficient sequence modeling is often evaluated with a fixed dimensional observation space $\mathcal{O} \in \mathbb{R}^d$ (Tay et al., 2021).

**Spatio-temporal tasks** can be modeled as a state space $\mathcal{S}$ with an observation space $\mathcal{O}$ that evolves over time according to dynamics model $\mathcal{D}$. In some domains, $\mathcal{O}$ is a variably-sized set, $\mathbb{O}_t$, that changes between environment instances, or over the duration of the sequence. This creates challenges in both concisely and effectively compressing a sequence of variably sized observations into some fixed dimensional latent representation of the state of the system $\mathcal{S}'_t := \mathbb{O}_{0...T}$. This can be exacerbated if the temporal duration of this task is indefinite as $T \in \mathbb{N}^+$.

As an example of this class of problem, StarCraft II (SC2) is a real-time strategy game where players build an economy and military in order to defeat the opposing team. Players are given imperfect information, they cannot view parts of the map outside of a line-of-sight dictated by their unit positions. Hence, the observation space of SC2 includes a variably sized set of units and buildings that enter and exit the player line-of-sight as the are built or destroyed. SC2 follows a typical rock-paper-scissors approach where there are counter-strategies that can be employed against a given player. Micromanagement in SC2 games plays an important role in the performance of a player, and analysis of this can give significant insight into strategy. Novices typically perform a few dozen actions per minute, whereas professional perform hundreds of actions per minute. These actions are typically taken on variably sized sets of units, buildings, targets or objectives.

---

[1]Environment and path-planning algorithm derived from here.

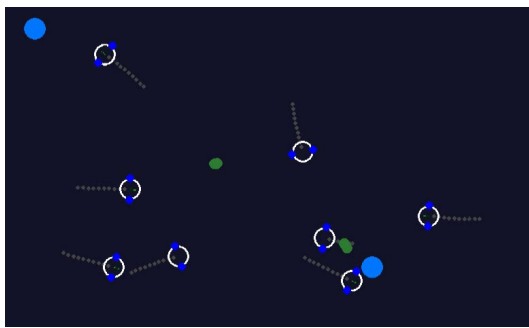

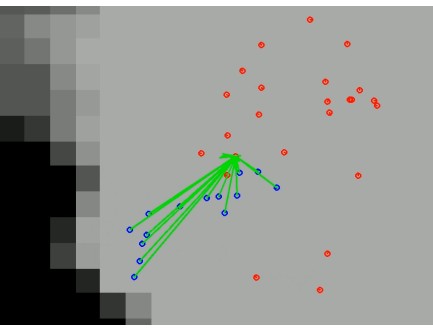

(a) Chasing-Targets gymnasium environment. Robots (marked with a trajectory trail) are randomly allocated targets (blue dots) to chase. When targets are reached (Green dots), robots are randomly assigned new targets.

(b) StarCraft II observation data. Blue circles are player units, red circles are enemy units, green arrows are unit-target assignments and the grayscale background is the height-map.

Figure 1: Visualisation of multi-agent environments used for benchmarking.

## 3 ENVIRONMENTS

Given the absence of suitable public time series datasets with irregular observations, we introduce a new set of intent recognition (Felip et al., 2022; Ahmad et al., 2016) benchmarks for evaluation. We describe these first, to provide context for the design motivations in Section 4. Each consists of two sets of agents which interact with one another with some objective. We use these environments to test whether models are capable of capturing relevant information from irregular observations of a complex environment, in order to infer some properties from it.

**The chasing targets environment** (Figure 1a) involves an arbitrary number of two-wheeled robots that chase an arbitrary number of particles bouncing around the environment, while trying to avoid colliding with one another. Each robot is randomly assigned a target particle at the beginning of the simulation and initialized in a random stationary pose. When the robot reaches the particle, it is randomly assigned to a new particle to chase. Robots are controlled with a simple cost function to select the best control inputs that minimizes the robot's distance to the target's projected position and a penalty term if a collision is forecast to occur with another robot. Particles are initialized at random positions and velocities. Here, the goal is to predict which target an agent is chasing, given observations of agent and particle positions and velocities. As a second, more challenging task, the observation space only includes the robots and the model must also predict where chased particles are located in the scene. This is challenging as not only the position of the particles are unknown, but the cardinality must also be estimated. This task is formulated as an occupancy problem, the model needs to estimate the likelihood that a position in the environment is occupied by a particle.

**The StarCraft II data** used is sampled from tournaments hosted between 2019 and 2023[2] and requires that we predict unit assignment actions based on prior observations of player and enemy units. To focus on battle sequences where players are micro-managing their units in combat, we target parts of the game when damage dealt or received by a player exceeds a threshold. Non-overlapping sequences of a fixed size are created at these instances. We note that observation data is irregularly sampled, hence the time duration of each sequence will vary. This domain has some subtleties compared to *Chasing-Targets* as units are not always assigned to an enemy unit. They can either be idle, or assigned to a position to move to. Hence, we introduce null option for the assignment problem. As a second more challenging task, we also consider the case where we are required to estimate if a unit has been given a target position command and the location of that target position. The assignment and the position estimation problem need to be jointly learned and performed by each model. The observation space for StarCraft II includes a terrain height-map and the position and properties (health, damage, etc.) of player and enemy units. The observation space is restricted to a region-of-interest (ROI) of a fixed size, see Appendix A.7 for detail on how the ROI is calculated. An example of an extracted ROI with unit data and their assignments is depicted in Figure 1b. Units outside of the ROI are truncated, resulting in a time-varying set of units in the

---

[2]Replays are sourced from https://lotv.spawningtool.com/replaypacks/

environment as they enter, exit or die in combat. The observability rules of the game apply from a player's point-of-view. Enemy units will be hidden in the fog of war or when an obfuscation ability is used, for example Zerg players can "burrow" units into the ground.

Both environments above are good examples of sequential modeling and prediction problems involving irregular, multidimensional time varying observation spaces. Below, we describe a set of efficient sequential encoding and dynamics modeling strategies suitable for these tasks.

# 4 METHOD

We evaluate several methods to encode spatio-temporal features into a series of latent states. The decoder for each of the tasks is fixed in order to isolate the contribution of the encoding methodology. Details on decoding methodologies is provided in the supplementary material (Appendix A.5).

## 4.1 SPATIAL ENCODING

This section outlines how the latent representation of the scene is constructed, using the StarCraft task above as an example. The scene observation is tokenized identically for each of these encoders. First, the $(x, y, \theta)$ pose of agents are sinusoidally encoded. Units from StarCraft II have additional information such as health, max health and a learned embedding representing the unit type. From this variably sized set of tokens, $\mathbb{O} = \{\boldsymbol{o}\}_{n_o}, \boldsymbol{o} \in \mathbb{R}^{d_o}$, where $n_o$ is the number of observed units, and $d_o$ is the feature dimension, we must summarize a fixed set $\mathbb{L} = \{\boldsymbol{l}\}_{n_l}, \boldsymbol{l} \in \mathbb{R}^{d_l}$ suitable for latent dynamics modeling.

We use two methods for summarizing $\mathbb{O}$ into $\mathbb{L}'$ (as illustrated in Figure 2), each of which involves MHA between $\mathbb{O}$ and $\mathbb{L}$. The first is a **BERT**-style (Devlin, 2018) transformer encoder. Latents $\mathbb{L}$ are concatenated to $\mathbb{O}$, and act as the "[CLS]" tokens of the BERT encoder. The encoded "[CLS]" tokens, $\mathbb{L}'$, are used as a fixed-size representation of the variably sized observation data. The second method uses a block of cross-attention layers (**X-Attn**), $\mathbb{L}$ to query key-value pairs generated from $\mathbb{O}$ to transfer relevant information. We denote these encoders as **Enc**.

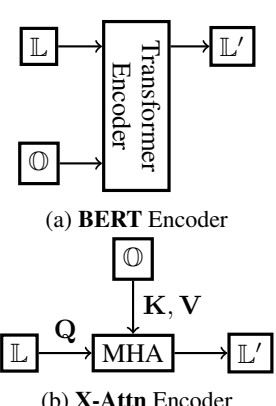

(a) **BERT** Encoder

(b) **X-Attn** Encoder

Figure 2: Encoders summarize an irregular set of tokens from the observation $\mathbb{O}$, to a fixed size $\mathbb{L}$, for the spatio-temporal encoder.

Without loss of generality, consider the case where we have two observation sources. In the tasks above, these correspond to the two teams of agents. Since there are two distinct observation sources, $\mathbb{O}_p$ and $\mathbb{O}_e$, we test a variety of methods to determine an effective method for combining both into $\mathbb{L}'$. The "Fused" method, Figure 3a, adds a learned embedding per source and then concatenates the sources together for encoding. This method has fewer parameters than the alternative methods as there is only one **Enc**. This also enables flexibility in gathering the optimal amount of information from each type of source. However, this comes at the cost of a larger attention matrix inside **Enc** with $\mathcal{O}((N_p + N_e)^2)$. The "Piece-wise" method, Figure 3b, encodes $\mathbb{O}_p$ and $\mathbb{O}_e$ separately with half of the latent state $\mathbb{L}$ used for each observation source. This method has the benefit of a smaller attention matrix $\mathcal{O}(N_p^2 + N_e^2)$, but enforces an equal weighting of information between the two sources, which is potentially sub-optimal. "Sequential" processes $\mathbb{O}_p$ then $\mathbb{O}_e$, as depicted in Figure 3c. This renders a smaller attention matrix, but reduces parallelism and increases the depth of the model.

The height-map is additional context provided in the SC2 task. ResNet-18 is used as the feature extractor, however we replace the final $1 \times 1$ adaptive-average-pool and fully-connected layers with a $4 \times 4$ adaptive-average-pool layer to create a grid of features. Sinusoidal position embeddings are added to the feature grid and then flattened, creating the set of contextual tokens. These tokens are appended to the extracted spatio-temporal features before being passed to the task decoder.

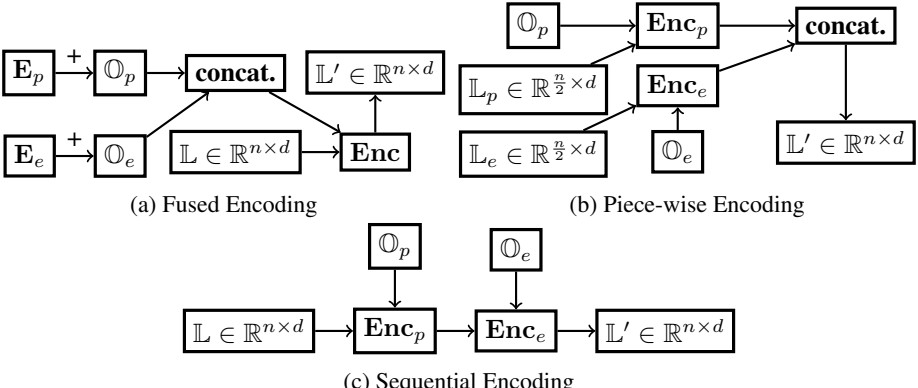

Figure 3: Several methods of encoding player ($\mathbb{O}_p$) and enemy ($\mathbb{O}_e$) observations to a fixed dimension $\mathbb{L} \in \mathbb{R}^{n \times d}$. Process together with an embedding to distinguish $\mathbb{O}_p$ from $\mathbb{O}_e$ (Fig. 3a), process separately (Fig. 3b) or process sequentially (Fig. 3c).

### 4.2 BASELINE SPATIO-TEMPORAL ENCODERS

We empirically evaluate a variety of models from categories mentioned in the Related Work (Section 7). As each of these models work on a 1D sequence, individual models process each token from the 2D latent state, with the exception of the **spatio-temporal transformer** (STT). We evaluate each of the **recurrent neural networks** included in PyTorch: **RNN**, **GRU** and **LSTM**. We find that a learned initial hidden state performs better than zero initialization (Appendix A.4). Hence, recurrent models use a learned initial state unless otherwise specified. We use **Mamba2** (Dao & Gu, 2024) to represent modern **SSMs**. For Mamba2 specific parameters, we use a state dimension of 64, convolution dimension of 4 and an expansion factor of 2.

We consider three transformer variants for temporal aggregation. The **spatio-temporal transformer** (STT) processes all the tokens from each time-step in one model. This method is similar to Agent-Former (Yuan et al., 2021), albeit without masking for "agent-aware attention". The **temporal-only transformer** (TT) uses individual transformers to process each latent. A learned embedding of the absolute time-step is added to the input tokens of the aforementioned transformers. We use **Ret-Net** (Sun et al., 2023) as a temporal encoder (no learned time-step embeddings) to represent the sub-quadratic family of transformers.

### 4.3 SCAN ENCODER

The key design objective of this spatio-temporal encoder is to efficiently aggregate historical information with a parallel algorithm, and to use that accumulated knowledge to resample the current observation. Furthermore, we use a set of tokens to represent our hidden state, rather than a single vector that is common to most algorithms. The motivation behind this is that attention mechanisms can be then utilized effectively with this hidden state, to either inject or extract information from this state. To achieve this, we use a weighted sum of an initially encoded input data from previous steps in the sequence. The accumulation can be performed efficiently in parallel over the sequence dimension with an inclusive scan operation (Merrill & Garland, 2016).

The proposed inclusive-scan algorithm uses the sequential algorithm (Fig. 3c) with X-Attn (Fig. 2b) as a recursive query driven sequential modeling approach. Inclusive-scan is performed on the updated variables to accumulate temporal information. When the observation is re-queried in the next layer, it is conditioned on an accumulated history. This process is depicted in Figure 4. For inference, we can calculate the next latent encoding with a scaled copy of the previous time-step $\mathbb{L}'_{xt} = \frac{1}{\gamma}\mathbb{L}'_{x(t-1)} + \mathbb{L}_{xt}$, resulting in $\mathcal{O}(1)$ memory and compute complexity with respect to sequence length. We show in Section 5 that cycling between cross-attention and inclusive-scan is superior to a block of cross-attention layers with an inclusive-scan at the end or sampling the input once, and cycling between inclusive-scan and self-attention. We include the number of layers as nomenclature for the scan encoder, for example Scan $4\times$ is a scan encoder with four cycles.

$$\mathbb{L}_0 \longrightarrow \boxed{\mathbb{L}_{1t} = \mathbf{Enc}_1(\mathbb{L}_0, \mathbb{O}_t)} \longrightarrow \boxed{\mathbb{L}'_{1t} = \sum_{i=0}^{t}(\tfrac{1}{\gamma})^{t-i}\mathbb{L}_{1i}}$$

$$\mathbb{O}_t \longrightarrow \boxed{\mathbb{L}_{2t} = \mathbf{Enc}_2(\mathbb{L}'_{1t}, \mathbb{O}_t)} \longrightarrow \boxed{\mathbb{L}'_{2t} = \sum_{i=0}^{t}(\tfrac{1}{\gamma})^{t-i}\mathbb{L}_{2i}} \longrightarrow \cdots$$

Figure 4: The inclusive-scan encoder alternates between sampling the observation based on some latent variable $\mathbb{L}_{xt}$ and accumulating a weighted sum $\mathbb{L}'_{xt}$, where $x$ and $t$ are the model layer and input sequence index respectively. $\mathbb{L}_0$ is an initial set of learned parameters and $\gamma \geq 1$.

An important subtlety is that the inclusive-scan is weighted so the historical contribution decays with $\gamma \geq 1$. This ensures that the accumulation does not diverge in magnitude over a long sequence. We show in Section 5 that the scan with $\gamma = 2$ outperforms $\gamma = 1$ (a simple cumulative sum). A PyTorch extension was written to efficiently perform the forward and backward method of discounted inclusive-scan on CPU and GPU with C++/CUDA[3].

## 5  RESULTS

We used PyTorch to train our models. Unless otherwise specified, experiments used a batch size of 64, AdamW optimizer with a learning rate of $1e^{-3}$ for *Chasing-Targets* and $1e^{-4}$ for *StarCraft II*, a polynomial schedule with power 0.9 and gradient clipping of $0.1$[4]. A two layer block of MHA is used for X-Attn and BERT encoders. Baseline temporal encoders also consist of two layers.

### 5.1  CHASING TARGETS

Each environment instance is randomly generated, sampling the number of robots and targets from a uniform distribution, randomly placing them on the field. The field is a $4 \times 4$m grid and the maximum velocity of the agents is 0.5m/s. The first 10 iterations of the simulation are skipped to remove the domain gap between the behavior of the robots after random stationary initialization, and the steady state chasing and switching targets. The duration of the simulation is 41 iterations.

#### 5.1.1  TARGET ASSIGNMENT

We train the target assignment challenge for $\approx 47$k iterations and use a latent state $\mathcal{L} \in \mathbb{R}^{8\times128}$. The number of robots in each simulation is sampled from $\mathcal{U}(8,15)$ and the number of targets $\mathcal{U}(3,6)$. We perform an ablation study on each of the encoding methodologies with the temporal transformer and find that the Sequential X-Attn performs the best (Figure 5). We include detailed results of the training cost and further discussion in Appendix A.1. From here on, Sequential X-Attn is used for comparing temporal encoders. We also find that a learned initial state for recurrent models generally performs better than states initialized as zero (Appendix A.4).

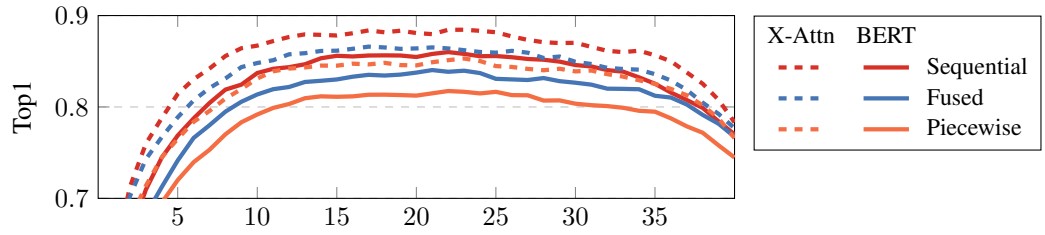

Figure 5: Encoding algorithm comparison with Temporal-Only Transformer (TT).

Figure 6 shows a mostly linear correlation between the accuracy and training cost (throughput, memory usage and parameter count) of each model. While Scan2$\times$ is the cheapest encoder to

---

[3]Link to source included upon acceptance, currently blinded for anonymity
[4]Link to source and model configurations will be provided upon acceptance

train, it has the poorest accuracy at 65% compared to the next model which is GRU at 73%. The most accurate encoder, temporal transformer (TT), outperforms second place by +6.5%. While it has the highest memory consumption and parameter count, the training step time is closer to the median. RetNet has a lower parameter count and memory usage compared to TT, however it is less accurate and its implementation is significantly slower. Scan $4\times$ doubles the number of layers, hence overall cost, however only observes a modest increase in performance +2.8%. This suggests stacking more layers suffers from diminishing returns. To attain better results, other modification should be performed. In 5.1.2, a self-attention layer is added after X-Attn to improve performance.

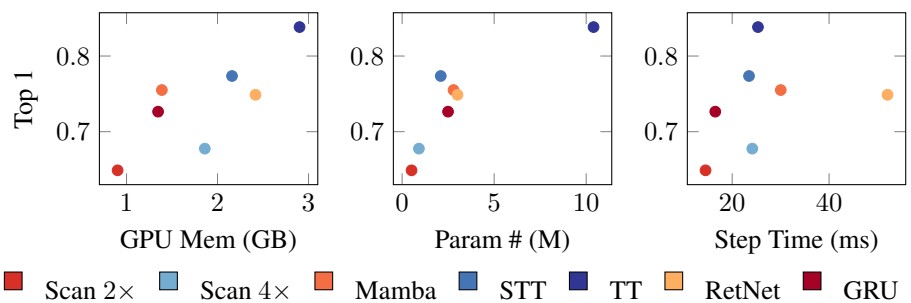

Figure 6: Average Top 1 assignment accuracy over the simulation sequence with training cost. The top left corner is ideal in each scenario.

We notice a trend of assignment accuracy decay after a peak earlier in the sequence in Figure 7. This is more pronounced in the better performing models and is correlated with the end of the sequence, rather than time of the sequence (Appendix A.1.4).

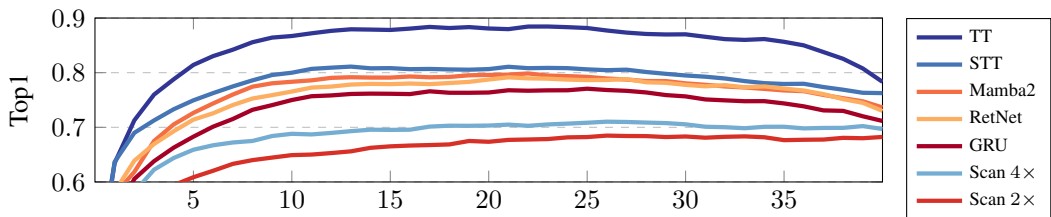

Figure 7: Assignment accuracy comparison over a sequence.

To understand the design contributions, we remove individual components from a Scan $4\times$ baseline (Figure 8). If a cumulative sum is performed without weighting, accuracy decays over time significantly after about 16 steps. Reducing the size of the latent state to $\mathbb{R}^{1\times128}$ has significant performance consequences as not enough historical information is retained over the sequence. This performs almost as poorly as not performing a scan at all, the worst performing variant. We evaluated more latent state sizes using the temporal transformer in Appendix A.1.

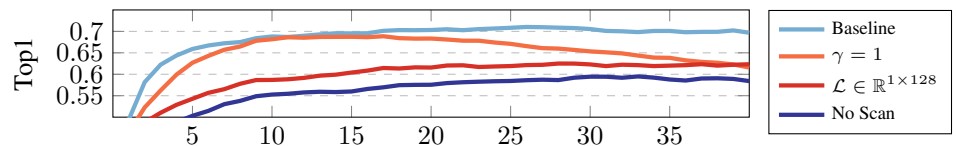

Figure 8: Contribution ablation for a Scan $4\times$ encoder over a sequence.

### 5.1.2 HIDDEN TARGET ESTIMATION

We trained the hidden target estimation challenge for $\approx 94$k iterations with a batch size of 32 and latent state $\mathcal{L} \in \mathbb{R}^{16\times128}$. The number of robots in each simulation is sampled from $\mathcal{U}(4, 12)$ and the

number of targets from $\mathcal{U}(2,5)$. Figure 9 shows that the Scan and Mamba encoders perform equivalently, whereas the transformer encoder fails on this task. Although AUC scores are numerically low, the models are often correctly able to infer target locations in the scene, as shown in Figure 10. While STT performed well in target assignment, it struggled with hidden target estimation.

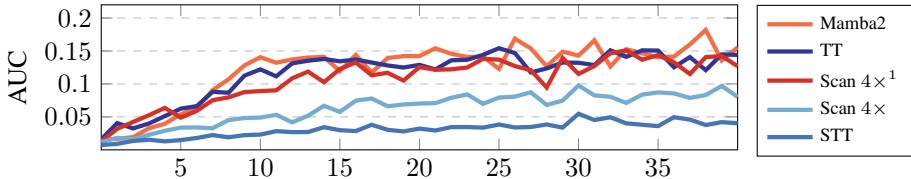

Figure 9: Model prediction AUC over the hidden target estimation sequence.

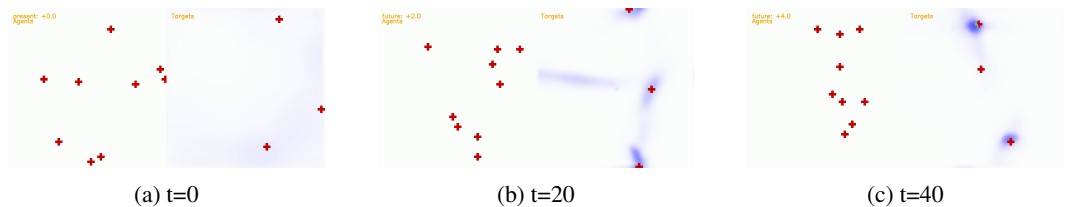

| (a) t=0 | (b) t=20 | (c) t=40 |

Figure 10: Rendered hidden target estimation over a 40 frame sequence where the left of each frame is the observable robots and the right is the hidden targets and model prediction. In the right frames, blue is the prediction and its intensity represents confidence, red pixels are the robots and targets.

## 5.2 REPLACING SCAN WITH GATING

We evaluated replacing the weighted sum operation in Scan $4\times$ with gating mechanisms, GRU and Gated Impulse Linear Recurrent (GILR) (Martin & Cundy, 2018). Our weighted sum is equivalent to a GILR that evenly weights the last hidden state and present value. Table 1 shows that accuracy is positively correlated with model sophistication at the expense of training cost. Unlike GRU, GILR can be parallelized over the sequence like our algorithm. While in theory GILR should have a lower cost than GRU, we expressed GILR as native PyTorch code. A CUDA implementation could fuse operations into a single kernel to improve throughput and reduce memory. We observed decaying accuracy at the end of the sequence, similar to Section 5.1.1. This decay is not correlated with sequence duration, as shown by evaluations on an 81 step sequence (Appendix A.1.4). Figures 13 and 14 show this phenomenon is related to the end of a sequence, not sequence length.

| Encoder | Top1 Acc. | # M params | Mem. GB | Train it/sec | Infer. it/sec |
|---------|-----------|------------|---------|--------------|---------------|
| Scan | 0.680 | **0.912** | **3.16** | **22.6** | **23.3** |
| GILR | 0.714 | 1.97 | 4.490 | 13.9 | 17.2 |
| GRU | **0.734** | 4.082 | 3.85 | 20.2 | 20.9 |

Table 1: Replacing the weighted sum of Scan $4\times$ with gating mechanisms on a 81 step sequence.

## 5.3 STARCRAFT II

We train each model for $\approx 205$k iterations with a sequence length of 30 and latent state $\mathcal{L} \in \mathbb{R}^{16 \times 256}$. StarCraft II is more challenging than *Chasing-Targets*, evidenced by the significantly lower top-1 assignment score for non-null assignments in Table 2. There are substantially more units on the field which vary in cardinality as they enter, exit and die in combat. Furthermore, in dense combat scenarios it is potentially more challenging to precisely assign targets. We find that a relative cartesian categorical representation of position performs the best (Appendix A.6) and is used hereafter.

While TT still performs the best across the board, its accuracy advantage over other models has diminished. Since the number of units is significantly greater in SC2 than *Chasing-Targets*, the

observation step becomes a greater component of the sequence processing cost. Furthermore, the sequence length of SC2 is $\approx 25\%$ shorter than *Chasing-Targets*. The greater training cost bias towards the observation stage reduces the proportional cost reduction using the Scan encoder. As the benefit of using a simpler encoder is diminished on a short sequence with complex observation space, taking advantage of a more complex encoder would likely be beneficial. Mamba2, LSTM and TT have greater accuracy than Scan3$\times$ at similar training cost. Importantly, all models are able to exploit the information provided by sequentially encoding irregular observations with X-Attn.

| Encoder | Top1 | Top1 +null | Top5 | MSE | F1 | # M Param. | Mem GB | Train it/sec |
|---|---|---|---|---|---|---|---|---|
| LSTM | 0.216 | 0.905 | 0.637 | 0.190 | 0.811 | 32.1 | **9.59** | 14.6 |
| TT | **0.258** | **0.918** | **0.647** | **0.169** | **0.844** | 57.4 | 10.6 | 14.7 |
| RetNet | 0.156 | 0.916 | 0.622 | 0.178 | 0.834 | 32.1 | 10.0 | 9.8 |
| Mamba2 | 0.238 | 0.913 | 0.643 | 0.180 | 0.832 | 29.1 | **9.59** | 14.3 |
| Scan 3$\times$ | 0.209 | 0.892 | 0.632 | 0.202 | 0.796 | **14.5** | 9.91 | **15.1** |

Table 2: Average performance over the SC2 battle sequence and training cost.

## 6  LIMITATIONS

Our benchmark sequences are performed over relatively short time-spans compared to other sequence modeling benchmarks (Tay et al., 2021). *Chasing-Targets* accuracy saturates early in the sequence, however we note accuracy decay in some encoders (Fig. 7). SC2 accuracy increases throughout the sequence. Micromanagement tasks may not elicit long range dependencies to the extent of short term ones, with players performing hundreds of actions per minute. There are a number of multi-agent interaction environments based on real-world observation data of vehicles (Ettinger et al., 2021; Wilson et al., 2021) and pedestrians (Robicquet et al., 2016). These benchmarks utilize shorter term prediction, often less than 10 seconds and $\leq 10$Hz resolution. Furthermore, they often only contain random short sequences (as opposed to a long sequence that is sub-divided for training and evaluating) or are limited in size as motion datasets require costly annotation of agents and their tracks. There may be additional value in the development and benchmarking of the models above on tasks with longer term temporal dependencies and irregular and complex observation spaces. However, we note that Auto-regressive SSM and RNNs are known to perform worse than Transformer and Linear models on long term forecasting benchmarks (Das et al., 2023). Assuming findings in Section 5.2 generalize to long-term forecasting, the Scan Encoder will likely perform similarly.

## 7  RELATED WORK

**State Space Models** (SSMs) like our approach are ubiquitous in sequential modeling tasks requiring efficiency. In control and dynamics modeling, they have been applied to learn locally linear latent dynamics models conditioned on exogenous inputs (Watter et al., 2015; Banijamali et al., 2018; Jaques et al., 2021; Levine et al., 2019; Karl et al., 2022; Han et al., 2020). More recently, these models have become popular with endogenous inputs. Here, SSMs parameterize a sequential process with endogenous input $\boldsymbol{u}(t)$ to model the output process $\boldsymbol{y}(t)$ with hidden state $\boldsymbol{x}(t)$,

$$\dot{\boldsymbol{x}}(t) = \boldsymbol{A}\boldsymbol{x}(t) + \boldsymbol{B}\boldsymbol{u}(t), \quad \boldsymbol{y}(t) = \boldsymbol{C}\boldsymbol{x}(t) + \boldsymbol{D}\boldsymbol{u}(t).$$

Parameters $(\boldsymbol{A}, \boldsymbol{B}, \dots)$ can be approximated in a discrete form $(\bar{\boldsymbol{A}}, \bar{\boldsymbol{B}}, \dots)$, appropriate for a sequence-to-sequence mapping, such that they can be learned with gradient descent. However, these matrices must be deliberately structured for effectiveness, usually guided by HiPPO theory (Gu et al., 2020). The key desirability of SSMs is re-parameterization as a convolution to enable parallel computation for training, $\boldsymbol{y} = \bar{\boldsymbol{K}} * \boldsymbol{u} + \bar{\boldsymbol{D}}\boldsymbol{u}$, where $\boldsymbol{K} = (\bar{\boldsymbol{C}}\bar{\boldsymbol{A}}^i\bar{\boldsymbol{B}})_{i=0\dots L-1}$ for sequence length $L$. Rather than learning the SSM matrices, Fu et al. (2023) propose learning $\bar{\boldsymbol{K}}$ directly with various regularization techniques and an efficient long convolution implementation to enable feasibility. Gu & Dao (2023) introduce the notion of time-dependent SSM matrices, similar to an LSTM, adding mechanisms for selectively propagating or forgetting state information dependent on the current token. This is also accompanied by a specialized implementation for performance. In follow up work, Dao & Gu (2024) decompose the SSM into diagonal and low-rank blocks for further efficiency.

**Sub-quadratic Transformer Attention** has been proposed with various techniques. **Linearized Attention** is one of the paradigms used to achieve this. Katharopoulos et al. (2020) re-parameterize multi-head self-attention with a non-negative feature map (i.e. $\text{elu}(x) + 1$) applied to $\mathbf{K}$, $\mathbf{Q}$ separately to attain linear time and memory complexity, and enable incremental processing in scenarios with causal masking. RWKV (Peng et al., 2023) utilizes learned position biases, similar to (Zhai et al., 2021), rather than queries from that position. Sun et al. (2023) follows a similar mechanism to our proposed method where a hidden state $\mathcal{S}$ is propagated with a decay factor $\gamma$ and updated with the current input, $\mathcal{S}_n = \gamma\mathcal{S}_{n-1} + \mathcal{X}_n$. However, our cyclic scanning and updating conditions the input on the accumulated history, before adding to the time-decaying latent state, as illustrated in Figure 4. **Blocked Attention** is another proposed method for avoiding $\mathcal{O}(L^2)$ complexity. However, a pure application of blocked attention would omit longer-term dependencies. To address this, Hutchins et al. (2022) introduce a set of recurrent states for long term context. Two components of the model run in parallel, one that leans how to gather context temporally ("horizontally"), and another than leans to extract long term information for the current decoding task ("vertically"). Didolkar et al. (2022) takes a similar approach, however interleaves long- and short-term information in their "Perceptual Module", the output of which updates the recurrent information.

**Irregular Spatio-Temporal Tasks** can be found in many domains, but there has been little to no focus on efficient temporal modeling with neural networks for this class of problems. Perhaps most relevant is multi-object tracking, which aims to associate detected objects between frames. Recursive processing is popular in end-to-end learning of the multi-object tracking problem (Meinhardt et al., 2022; Zhang et al., 2023; Zhu et al., 2022), but this is only suitable when training short-term tracking algorithms on benchmarks with much shorter timescales (Dendorfer, 2020; Dave et al., 2020) than those proposed above. Over a longer horizon, for example in long-term Re-identification tasks, these recursive methods scale poorly, hence further innovation is required. Longer horizon methods (Qin et al., 2023) are typically are not trained end-to-end for this reason. Graph neural networks (GNNs) (Battaglia et al., 2018) are popular architectures for variable dimensional or irregularly structured data due to their permutation invariance. However, these models generally rely on fixed graph structures for global context, although some approaches attempt to model graph evolution graph over time (Pareja et al., 2020; Manessi et al., 2020). This lacks the scalability of state space models and GNNs are difficult to frame in an incremental manner for inference, typically processing the full sequence. For example, Gao et al. (2020) repeatedly construct a graph representation of traffic scenes with agent tracklets and environment context to produce future trajectory estimates.

## 8 CONCLUSION

This work introduces a sequential modeling approach for irregular observation sources of varying dimensionality and cardinality. We investigate the use of various encoders to pack these observations into a fixed dimensional latent, and corresponding sequence modeling approaches. We propose a novel algorithm that alternates between cross attention and a weighted inclusive scan to accumulate context over time and show that this performs on par with existing sequence modeling approaches on two new challenging benchmarks, with comparably higher training and inference speeds and a lower parameter count. A key benefit of the proposed approach is that it can naturally be tweaked to allow a broad range of performance/compute trade-offs (see Appendix A.3 for more extensive experiments in this vein) as required for a given application.

## 9 REPRODUCIBILITY

The dataset, simulation, source code and experiment configuration files are publicly available, but currently not linked for anonymity.

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

# A APPENDIX

## A.1 ENCODER ABLATIONS

### A.1.1 SPATIAL ENCODER RESULTS

We evaluate the performance characteristics of each encoding method using the temporal-only transformer on the *Chasing-Targets* assignment task. From Table 3, we find the following orders of accuracy holds, sequential > fused > piecewise, and **X-Attn** > **BERT**. In this case, X-Attn demonstrates to be a clear winner over BERT, with using less memory, faster inference time and greater accuracy. Hence, empirically, we find that a sequential encoding with **X-Attn** is the best performing algorithm in terms of accuracy. While the BERT fused method produces a larger attention matrix

| Encoder | Algorithm | Top1 Acc. | # M params | Mem. GB | Train it/sec | Infer. it/sec |
|---------|-----------|-----------|------------|---------|--------------|---------------|
| BERT | piecewise | 0.769 | 10.4 | 2.88 | 38.8 | 62.7 |
| BERT | fused | 0.789 | **9.99** | 2.98 | 44.4 | 66.5 |
| BERT | sequential | 0.809 | 10.4 | 3.64 | 35.5 | 59.3 |
| X-Attn | piecewise | 0.805 | 10.4 | 2.44 | 41.7 | 58.5 |
| X-Attn | fused | 0.820 | **9.99** | **2.19** | **46.2** | **71.3** |
| X-Attn | sequential | **0.838** | 10.4 | 2.90 | 39.4 | 53.3 |

Table 3: Accuracy and cost of spatial encoders for *Chasing-Targets* with Temporal Transformer.

$\mathcal{O}((N_L + N_a + N_t)^2)$ compared to the sequential method $\mathcal{O}((N_L + N_a)^2 + (N_L + N_t)^2)$, where $N_L, N_a, N_t$ are the number of latent variables, agents and targets respectively, we find the fused method uses considerably less memory. The additional parameters and doubling of layers associated with the sequential algorithms seem to outweigh the larger attention matrix of the fused method. Furthermore, the fused method is consistently faster. This is expected compared to the sequential model, however not for the piecewise which can independently process the target and agent branches in parallel. This is likely due to PyTorch not actually processing the piecewise operations in parallel. To enable processing of piecewise operations, new CUDA streams need to be manually invoked and correctly managed[5]. The piecewise algorithm is consistently inferior to the other methods in accuracy, indicating that enforcing an even weighting between information from agent and target observations is suboptimal.

### A.1.2 LATENT STATE DIMENSIONS

We performed an ablation on the latent state size of the X-Attn sequential temporal-only transformer (Figure 11). We find that the optimal size is $8 \times 128$, more tokens has diminishing returns, and less tokens or dimension has reduced accuracy.

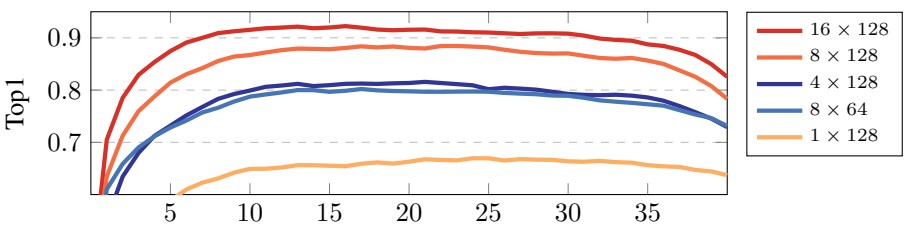

Figure 11: Top1 Accuracy of X-Attn Sequential Temporal-Only Transformer on *Chasing-Targets*

### A.1.3 GAMMA ABLATION

We briefly evaluate a range of $\gamma$ factors to validate that $\gamma = 2$ performs the best (Figure 12). $\gamma = 1$ is a simple unweighted cumulative sum, results in severe accuracy degradation over time. The other

---

[5]https://pytorch.org/docs/stable/notes/cuda.html#cuda-streams

two tested factors $\gamma = 1.5, 3$ are slightly worse than $\gamma = 2$, however we would expect values $> 3$ to further perform poorly.

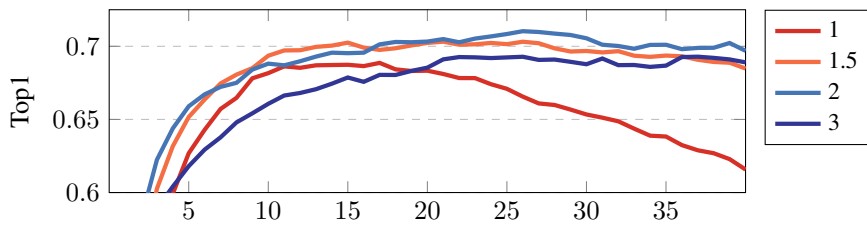

Figure 12: Top1 Accuracy of Scan Encoder with different $\gamma$ factors

### A.1.4 REPLACING SCAN WITH GATING

We observed that the accuracy of the gated models decayed towards the end of the sequence (Figure 13). We believe that this phenomenon is correlated with it being the end of the sequence, not correlated to the length of the sequence itself. We confirmed this by running an experiment with a longer duration in Figure 14, also showing the same accuracy decay phenomenon, invariant of the actual duration of the sequence. The GILR implementation was based on Heinsen (2023).

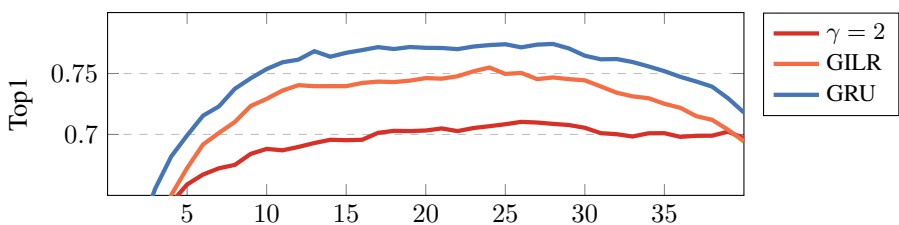

Figure 13: Comparison between weighted sum, MinGRU and GRU.

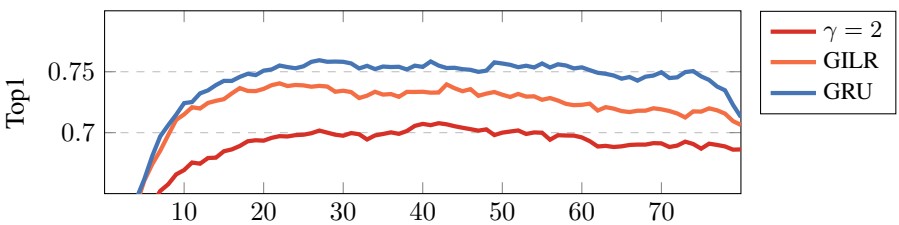

Figure 14: Comparison between weighted sum, MinGRU and GRU over an extended sequence.

### A.2 OCCUPANCY ESTIMATION TRAINING COST

Table 4 shows the occupancy decoder uses substantially more memory due to rendering a dense $64 \times 64$ occupancy map, rather than correlating at most 15 robots to 6 targets. We improve the representation capability of Scan $4\times$ by introducing an additional self-attention layer after the X-Attn. This results in Scan $4\times$ approaching the accuracy of the STT and Mamba2 encoders, while still using significantly fewer parameters with similar memory and compute.

### A.3 CHASING TARGETS DETAILED RESULTS

Table 5 provides a larger set of results including various encoder (BERT, X-Attn) and encoding algorithm (Fused, Piece-wise) variants for the S.T, Recurrent and Mamba2 encoders. We also include

| Encoder | AUC | Soft IoU | # M param. | Mem. GB | Train it/sec | Infer. it/sec |
|---------|-----|----------|------------|---------|--------------|---------------|
| STT | 0.031 | 0.008 | 1.702 | 13.1 | 11.9 | 12.6 |
| Mamba2 | **0.122** | **0.031** | 4.284 | 11.0 | 12.0 | 12.2 |
| TT | 0.116 | 0.028 | 19.49 | 13.1 | 13.0 | 14.2 |
| Scan 4× | 0.063 | 0.014 | **0.514** | **10.9** | **15.1** | **15.1** |
| Scan 4×[1] | 0.106 | 0.026 | 0.912 | 12.3 | 13.0 | 13.0 |

Table 4: Accuracy and training cost of each temporal encoder. [1] notes an additional self-attention layer in the model.

| Encoder | Top1 Accuracy | | | | # M param. | Mem. GB | Train it/sec | Infer. it/sec |
|---------|------|------|------|------|------------|---------|--------------|---------------|
| | 0 | 5 | 10 | 40 | | | | |
| RNN-L,Fuse,BERT | 0.359 | 0.529 | 0.592 | 0.633 | 0.76 | 1.82 | 60.7 | 89.4 |
| GRU-L,Fuse,BERT | 0.372 | 0.652 | 0.706 | 0.711 | 1.29 | 1.88 | 58.8 | 86.6 |
| LSTM-L,Fuse,BERT | 0.375 | 0.646 | 0.703 | 0.703 | 1.56 | 1.89 | 56.4 | 81.4 |
| STT,PW,BERT | 0.377 | 0.688 | 0.739 | **0.719** | 2.10 | 2.81 | 35.3 | 55.2 |
| STT,PW,X-Attn | 0.374 | **0.692** | **0.745** | 0.716 | 2.10 | 1.86 | 42.7 | 52.1 |
| STT,Seq,X-Attn | 0.374 | **0.692** | **0.745** | 0.716 | 2.10 | 1.86 | 42.7 | 52.1 |
| Mamba2,PW,X-Attn | 0.381 | 0.673 | 0.724 | 0.703 | 2.79 | 1.04 | 30.9 | 31.6 |
| Mamba2,Fuse,BERT | 0.386 | 0.667 | 0.719 | 0.713 | 2.79 | 1.95 | 35.4 | 61.0 |
| Mamba2,Seq,Attn | 0.366 | 0.726 | 0.783 | 0.736 | 2.79 | 1.39 | 33.3 | 49.0 |
| Scan 1× | 0.354 | 0.479 | 0.521 | 0.576 | 0.313 | 0.61 | **104.0** | **125.6** |
| Scan 2× | 0.346 | 0.609 | 0.649 | 0.682 | 0.512 | 0.90 | 68.9 | 79.1 |
| Scan 4× | 0.360 | 0.659 | 0.688 | 0.697 | 0.912 | 1.86 | 41.4 | 43.8 |
| Scan 6× | **0.392** | 0.657 | 0.696 | 0.690 | 1.311 | 2.05 | 30.2 | 31.6 |

Table 5: Algorithm performance, training and evaluation cost of tested algorithms on *Chasing-Targets*.

slight variations of the Scan encoder with removed self-attention layers to trade accuracy and efficiency (indicated by [1]) or replace cross-attention with the observation in the repeat layers, with self attention blocks (indicated by [2]).

## A.4 RECURRENT MODEL COMPARISON

We evaluated all the available recurrent neural network modules available in PyTorch, comparing an initial hidden state which is a learned parameter or zeros. Each of these models use the **X-Attn** encoder with sequential encoding algorithm. We plot the accuracy of each model over time in Figure 15 and summarize the average accuracy and the training cost in Table 6. The standard RNN performs poorly whilst the LSTM and GRU are similar. While there is only small difference between the learned and zero initial hidden state for each model, the learned initial state has a slight advantage for both the RNN and GRU. From these results, the GRU stands out as a compelling choice since its accuracy exceeds the more complex LSTM, while costing closer to the RNN in training throughput and memory. However we observe a strange outlier in memory consumption of the GRU with zero initialization.

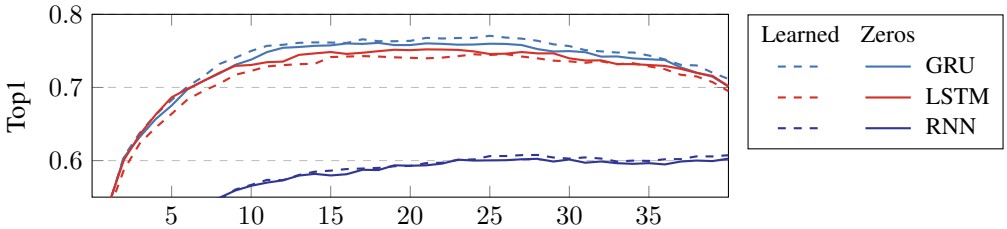

Figure 15: Comparison between recurrent models and hidden state initialization.

| Encoder | Init. | Top1 Acc. | # M Param. | Mem. GB | Train it/sec | Infer it/sec |
|---------|-------|-----------|------------|---------|--------------|--------------|
| RNN | Zero | 0.569 | **1.44** | **1.31** | 62.2 | **69.1** |
| RNN | Learn | 0.571 | **1.44** | 1.33 | **62.1** | **69.1** |
| GRU | Zero | 0.720 | 2.49 | 1.54 | 59.5 | 64.9 |
| GRU | Learn | **0.726** | 2.49 | 1.38 | 60.6 | 65.7 |
| LSTM | Zero | 0.715 | 3.02 | 1.46 | 55.5 | 62.1 |
| LSTM | Learn | 0.707 | 3.02 | 1.49 | 55.5 | 61.4 |

Table 6: Average Top1 Accuracy and Cost of Recurrent Models on *Chasing-Targets*

## A.5 TASK DECODING

In this section, we detail the decoding strategy used for each benchmark task. The foundation of each task decoder is a cross-attention operation between the ego-agents and latent state to gather the temporally aggregated observation information,

$$\mathbb{O}'_p = \textbf{X-Attn}(\textbf{Linear}(\mathbb{O}_p), \mathbb{L}'),$$

where the query for MHA is derived from the first argument of **X-Attn**, and the key-value from the second argument. We note that **X-Attn** and **Linear** are learned for each task.

**Goal Assignment** aims to find the correlation between agents and targets. We achieve this by learning a projection of the targets which can be correlated with the agent features,

$$\mathbb{O}'_e = \textbf{Linear}(\mathbb{O}_e), \quad \mathbb{A} = \textbf{CosineSimilarity}(\mathbb{O}'_p, \mathbb{O}'_e).$$

For the StarCraft II challenge, a learned null token is appended to the set of projected targets. Finally, the correlation between agents and targets is determined using a dot-product. Soft-max is applied along the target dimension to calculate the final assignment score from agent to target. We note that this is a one-to-many problem, many agents can have the same target, but each agent only has one target, including a null assignment.

Negative log likelihood (NLL) is used as a categorical loss between the predicted and the ground truth assignment. However, there are some complications due to the variable agents and targets in the scene. Invalid agents can simply be omitted from the loss function with a mask. To handle invalid targets, we group agents with the same number of targets in the scene and truncate the tensor to this number. This allows us to apply NLL loss while avoiding gradients from invalid targets and agents. In the StarCraft II task, there is a significant imbalance between agents with a target and those without. A weighting factor of $0.05$ is applied to emphasize non-null assignment losses.

A **Movement Target Assignment** task completes the motion action space that units in StarCraft II can take. Here, the objective is to estimate the coordinates on the map where the unit is moving to. This plays an important part of the game as maneuvering units into advantageous positions such as the high ground or choke-points is an important strategy in game-play. We evaluate several methods of estimating the target position with both regression and categorical techniques including: global cartesian, relative cartesian and relative polar. The difference between these methods is the number of channels of the final linear layer and the loss used for training. Furthermore, a logit is emitted to estimate the likelihood that the unit is following a position command and position estimate is valid.

L2 loss is used for regression-based position estimation and HL-Gauss (Imani & White, 2018) is used for categorical-based position estimation. We include additional logic in the HL-Gauss loss to correct angle wrapping for polar coordinate estimation (Appendix A.6). Binary cross-entropy loss is used for the valid position command logit. Position commands are under-represented in the dataset compared to target commands. To increase the amount of position data, we create pseudo-position commands from target unit coordinates while treating the position logit as false.

**Hidden Target Estimation** in *Chasing-Targets* is formulated as an occupancy prediction problem. An array of coordinates are sinusoidally encoded, $\mathcal{P}(.)$, to query the temporal encoding,

$$\mathcal{O}(x, y) = \textbf{Linear}(\textbf{X-Attn}(\text{concat.}(\mathcal{P}(x), \mathcal{P}(y)), \mathbb{L}')),$$

to yield likelihood of hidden target occupancy $\mathcal{O}$ at the position $(x, y)$. Focal loss (Lin et al., 2020) is used to address class imbalance between unoccupied and occupied pixels. Although position queries are continuous, for simplicity we use a fixed grid.

## A.6 STARCRAFT II UNIT TARGET POSITION REPRESENTATION

HL-Gauss (Imani & White, 2018) loss is used for categorical position encoding. Instead of a one-hot encoding label, a gaussian kernel is used as the categorical target with $\mu$ equal to the position and $\sigma$, which we set to $0.2$. The appropriate number of bins used for the prediction is correlated to the $\sigma$ chosen (Farebrother et al., 2024), we use 20. We decode the categorical output of the model by integrating over the product of the soft-max output and the value each bin represents. We perform an ablation study with the large Scan $6\times$ encoder, shown in Table 7, to determine the best performing position learning schema. Based on these results, we use a relative cartesian coordinate system for unit position estimation, encoded with a categorical distribution.

| Rep. | Frame | Encoding | MSE | | | |
|------|-------|----------|-------|-------|-------|-------|
|      |       |          | 0 | 5 | 15 | 25 |
| Cart. | Rel. | Scalar | 0.300 | 0.258 | 0.253 | 0.251 |
| Cart. | Glbl. | Scalar | 0.299 | 0.259 | 0.254 | 0.241 |
| Polar | Rel. | Cat. | 0.258 | 0.221 | 0.216 | 0.216 |
| Cart. | Rel. | Cat. | **0.248** | **0.211** | **0.207** | **0.206** |
| Cart. | Glbl. | Cat. | 0.254 | 0.220 | 0.218 | 0.219 |

Table 7: Position representation ablation performed with Scan $6\times$ encoder. Here Cart. is cartesian, Rel. is relative position from the unit, Glbl. is global position in the ROI and Cat. is categorical position encoding.

Using a polar coordinate system requires consideration for correct behavior of wrapping $\theta$, especially for a categorical representation. To create the categorical target, we double the template range to $[-4\pi, 4\pi]$. Once the kernel is projected onto the template, we add the reflection of the additional range to the opposite side in the normal range, i.e. we reflect and add the bins from $[-4\pi, -2\pi]$ to $[0, 2\pi]$, and repeat for the the opposite tail. To handle angle wrapping for decoding, we find the argmax category and rotate the categories to center on the argmax. With this new categorical range, the value that each bin represents is recalculated to the new range. The standard decoding algorithm is used to yield the final result. We test this algorithm by encoding and decoding a uniform range of targets $[-2\pi, 2\pi]$ and checking the original target is reconstructed.

## A.7 STARCRAFT II ROI CALCULATION

The objective of the ROI is to contain the main action happening on the screen, namely the units in combat. To achieve this, we gather the coordinates of all the units with target or motion commands over the sequence. K-Means clustering is performed on the positions and sorted by the number of members in the clusters. If the limits centroids of the K-means clusters can fit within the desired ROI size, then the mean of the centroids is used for the center of the ROI. If this condition is not fulfilled, the smallest cluster is removed. This is done iteratively until the condition is satisfied. This calculation can be performed live in the dataloader during training, or calculated once and saved a key associated with the sample to be re-indexed during training.

