# OpenReview forum: "Efficiently Scanning and Resampling Spatio-Temporal Tasks with Irregular Observations"
_ICLR.cc/2025/Conference — Submitted to ICLR 2025_

### Official Review · Reviewer_7WZF · 2024-11-01

**Soundness:** 3
**Presentation:** 2
**Contribution:** 2
**Rating:** 6
**Confidence:** 3

**Summary:**

The paper tackles the issues of spatio-temporal modeling in the context of variable observation spaces, such as when there are variable objects across episodes / scenes, etc or when the object number varies within an episode (such as when entities enter and leave an arena). The authors introduce a few baseline tasks representative of this problem and cover various attention-based encoding and temporal aggregation schemes, some of which are advertised as being novel (such as some of the encoder methods). They perform various analysis on different methodologies, such as task performance, memory, speed, etc on these tasks which highlight some of the benefits of various methods (such as the scan-based one).

**Strengths:**

Overall the paper is interesting and approaches a very important problem I think in a comprehensive way, covering some benchmarks that seem well motivated to me as well as some methodologies for dealing with the variable observation problem. The experiments I think are clear enough and the conclusions drawn I think are sound, demonstrating that some methods perform better than others.

**Weaknesses:**

There are places where the writing is good, and there are places where the writing and motivation is not so good. The introduction is very vague and lacks citations on some critical claims in the paper that make it difficult for readers who are not very familiar with the domain to understand well. The benchmarks are introduced well enough, but it's unclear why we need multi-player problems to address the core problem of variable observation spaces. Critically though when the methods come around the organization of the paper suffers, as it becomes extremely difficult to understand the motivation of different design choices, such as the scan operation (e.g., why was the scheme in figure 4 motivated, other than post-hoc performance?). The methods in Figure 3 seem interesting, but I had difficulty finding their consequence in the paper later on.

(Many of these and questions below were addressed sufficiently in the rebuttal and revision)

**Questions:**

1) Could you clarify the contributions of the paper in terms of the models introduced, notably wrt to SSMs. I read the related works, but they did not do a good job connecting to what was done in prior sections.
2) Why is it important that the benchmarks include PvP / PvE games?
3) Did you ablate over decoders when you decided on the final scheme to compare methods? Is it possible that some results are decoder dependent (ie there's a relationship between decoder and encoder results).
4) Could you explain in more detail the motivation of the scan methods? What do you think is going on that makes this method work? What does this have to do with variable observation spaces?
5) What is Lx in 4.1.1? What does the first subscript mean in L_xt?
6) Does the model ever output observation predictions? You mention "sample the current observation" but it's unclear what you mean.
7) What do you mean by "learned initialization" in 4.1.2?
8) What is the consequence of those PvE encoders from Figure 3 on the results?

---

> ### Author Response · Authors · 2024-11-26
>
> > There are places where the writing is good, and there are places where the writing and motivation is not so good. The introduction is very vague and lacks citations on some critical claims in the paper that make it difficult for readers who are not very familiar with the domain to understand well.
>
> We have added citations to relevant works in the introduction as well as tweaks to the main body and list of contributions to reflect the additional studies we have done in this revision and improve the quality of writing.
>
> > The benchmarks are introduced well enough, but it's unclear why we need multi-player problems to address the core problem of variable observation spaces.
>
> While our approach is applicable across a broad set of domains, these multi-agent problems are particularly representative of the hardest aspects of the irregular observation problem, going beyond say missing data imputation in tabular data. We note that in multi-player, multi-unit games like Starcraft, the observations can be considered a tree-like data structure, with each player representing a branch, and units and their attributes sub-trees in the scene. When observations are not available, or units appear or disappear, the tree structure changes quite dramatically. Existing motion datasets are usually brief or small due to manual annotations required. Multi-agent modelling problems like those chosen are good examples of spatio-temporal modelling tasks with irregular observations, and also of interest in the machine learning community.
>
> > Critically though when the methods come around the organisation of the paper suffers, as it becomes extremely difficult to understand the motivation of different design choices, such as the scan operation (e.g., why was the scheme in figure 4 motivated, other than post-hoc performance?).
>
> > Could you clarify the contributions of the paper in terms of the models introduced, notably wrt to SSMs. I read the related works, but they did not do a good job connecting to what was done in prior sections.
>
> Our results primarily set out to explore the trade-offs between efficiency and performance in the irregular observation case. This means results should be interpreted given the compute budget available to a practitioner. If only performance matters, practitioners should select a temporal transformer with sequential cross attention (X-Attn Sequential) for observation encoding. For efficiency and model parsimony, an inclusive scan should be selected with X-Attn Sequential. We have made this clearer in the text, and by adding an additional experiment where we directly replace the weighted cumsum with GRU or GILR. This experiment shows a more direct correlation of accuracy with temporal model sophistication and cost. We also changed to a Scan 3\times model in SC2 which now has the best training throughput, median memory usage with slight disadvantage in accuracy.
>
> > The methods in Figure 3 seem interesting, but I had difficulty finding their consequences in the paper later on.
>
> We have improved the motivation and clarity around the choice of encoders and dynamics models. We conducted an ablation study on all the spatial encoding methods with a fixed temporal encoder and found X-Attn Sequential to have the highest accuracy, while X-Attn Fused was the cheapest. For consistency, we now use the X-Attn Sequential algorithm for all the models in the paper.
>
> > Why is it important that the benchmarks include PvP / PvE games?
>
> This is a challenging problem as targets are often switched during the sequence, requiring the model to be adaptable to instantaneous changes in the signal. This is a potential reason why an RNN fails at this task.
>
> > Did you ablate over decoders when you decided on the final scheme to compare methods? Is it possible that some results are decoder dependent (ie there's a relationship between decoder and encoder results).
>
> Yes, it is possible that some results are decoder dependent. We aimed to reduce this by evaluating the encoding methods with relatively simple intention prediction tasks, rather than more difficult tasks such as long horizon forecasting or using RL to train an agent on the task.
>
> The main decoding ablation was the position decoding scheme. The target assignment can be seen in a similar vein to CLIP, correlating words and images, which is why we choose cosine-similarity of encoded agent and target features, although we have a one-way assignment which allows for a row-wise softmax.
>
> TBC.

---

> ### Author Response · Authors · 2024-11-26
>
> Cont.
>
> > Could you explain in more detail the motivation of the scan methods? What do you think is going on that makes this method work? What does this have to do with variable observation spaces?
>
> Our results primarily set out to explore the trade-offs between efficiency and performance in the irregular observation case. A parameter-free inclusive scan is the cheapest possible method to accumulate the spatial-encoding over the sequence dimension. We hypothesize that embedding the notion of recency in the latent with the discount factor has enabled better model adaptation of target assignment changes throughout the sequence, which is why a standard RNN is significantly worse than our Scan method or a GRU, GILR or LSTM. We show in A.1.2 that a larger latent state improves performance, the parameter-free Scan method avoids a corresponding proportional increase in parameter count and can be performed efficiently in parallel.
>
> > What is Lx in 4.1.1? What does the first subscript mean in L_xt?
>
> We have updated the paper to explicitly note that x denotes the layer index of the model.
>
> > Does the model ever output observation predictions? You mention "sample the current observation" but it's unclear what you mean.
>
> We make predictions on properties of the environment that aren’t observed by the model. Since we use multi-head attention, with exception of the first layer, since we have accumulated knowledge from the scan operation, our next query includes historical features, so the attention on the key generated by the observation is now conditioned on the history.
>
> > What do you mean by "learned initialization" in 4.1.2?
>
> The initial state of the recurrent neural network, we have adjusted the language to improve clarity.
>
> > What is the consequence of those PvE encoders from Figure 3 on the results?
>
> We conducted an ablation study (A.1.1 and Fig. 5) on all the spatial encoding methods with a fixed temporal encoder and found X-Attn Sequential to have the highest accuracy, while X-Attn Fused was the cheapest. For consistency, we now use the X-Attn Sequential algorithm for all the models in the paper unless otherwise specified.

---

> > ### Comment · Reviewer_7WZF · 2024-12-02
> > **Response to rebuttal**
> >
> > OK, if I'm to summarize some of the rebuttal to some of the more sticky points of my review here:
> > 1) Parameter-free inclusive scan was chosen as a minimally viable example of a model that has capabilities to accumulate spatial encodings, and RNN-based methods perform substantially worse.
> > 2) The consequence of methods introduced in Figure 3 have been clarified with an ablation provided that reveal a tradeoff between performance and computational cost.
> > 3) The motivation of using multiplayer games as irregular observation benchmarks has been clarified through its structure. I do agree that these games represent suitable benchmarks for the problem at hand. But I do think though a better job could be done in the presentation to connect the problem at hand (irregular observations) and these specific benchmarks. Perhaps it would be better to (in sections 2 and 3) specifically characterize different facets of the problem (via enumeration), then introduce the benchmarks and things that satisfy some or all of these facets. Currently, these specifications are woven through the first 3 sections, which ultimately leads to a poor setup of motivation of the environments in 3.
> > 4) Citations and other clarifying statements were made in response to my concerns.
> >
> > I am watching some of the discussion happening in rRfx. I do think that the latest draft is more or less sufficient wrt studying the effects of architecture.
> >
> > Given the above I will raise my score to 6.

---

> > > ### Author Response · Authors · 2024-12-03
> > >
> > > Thank you for your feedback and updating your score.

---

### Official Review · Reviewer_rRfx · 2024-11-02

**Soundness:** 3
**Presentation:** 2
**Contribution:** 2
**Rating:** 6
**Confidence:** 3

**Summary:**

The paper proposes a method for sequential prediction tasks that can handle observations of varying sizes at different times. It also simultaneously introduces two intention-prediction tasks in which to test this method. Experiments are performed against a few baselines and components of the method are ablated.

**Strengths:**

- The contribution of the task based on StarCraft can be an interesting domain for future work to study
- The writing is in general fairly clear to follow

**Weaknesses:**

I should start off by pointing out that I'm not precisely acquainted with this literature, so that will affect my judgment.

## Major points
1. One large issue I see with the paper is that I'm not totally certain why these particular tasks are good ones in order to explore modeling arbitrary-sized observation spaces. I think StarCraft as a domain makes sense, but there has been much prior work in StarCraft (which is not cited in the paper) that has used it in a different way, and I don't understand why the decision was made to couple "intention-prediction" tasks with the method. It seems that some domains like sequential decision making in healthcare, or some kind of variable-sized text prediction task would be equally fine as a benchmark domain. This is further complicated by the relatively unclear definition of the exact input and output for each task in Section 3. Given the motivation of these tasks is that it shows up often in interactions, it would seem that a space like Shared Autonomy might provide some more common benchmarks.

2. Another big issue that I see is that the paper does not make a strong enough case for the choice of baselines. I would expect there to be accompanying work in the domain previously with which to compare against, but because 4.1.2 does not cite any prior work, it's unclear to me that these choices are well-founded. Certainly I'm familiar with Mamba2, but the rest is unclear. Also the choice of encoders for the RNN methods feels somewhat unjustified.

3. The last major issues I see is that the results in Figure 5 and Table 1 feel somewhat inconclusive. Because nothing is controlled (either parameter count, memory, or training time) between methods, and there is no clear winner across metrics, it's difficult to see the benefit. When coupling these mixed results with the choice of new bespoke domains, it's even harder to understand the longer-term contribution. I don't think it's the case that a new method needs to be always better than prior work, but the fact that it does not appear to have particular advantages in computation that was claimed as a contribution ("efficiently address sequence modeling" at the end of Section 1), makes it hard to understand the contribution.

## Specific points

1. Section 4.1.2 add a citation to Mamba2
2. Section 4.2 it would be nice to have the problem written out mathematically as I find the current description a bit hard to follow
3. Section 5 is missing comment on hyperparameter tuning, especially for baselines
4. Fig 7 should be referenced in 5.1.1

**Questions:**

1. Section 3.2 Where exactly does the StarCraft data come from? It would be nice to have a detailed description of the data collection and formatting process in the Appendix
2. How different is this method of encoding observations from previous work on StarCraft (regardless of the exact task, the encoding stage should be similar).
3. What is the "learned initialization" in Section 4.1.2
4. Is there some reason why baselines couldn't be matched on some axis? (parameter count is probably easiest)
5. What I understand in Section 3.2 is that the StarCraft task is a 3-class (idle, move to position, move to enemy) classification problem given sequential context, but still this doesn't exactly specify the setup for me, could you clarify what exactly is the task?
6. Section 7 Given the related work (especially Sun et al. 2023) is there some reason why none of these were used as baselines?
7. The limitations seem a bit unfortunate, but also foreseeable, is there some reason that the authors specifically designed their own benchmarks instead of working with something longer-term?

---

> ### Author Response · Authors · 2024-11-26
>
> >  One large issue I see with the paper is that I'm not totally certain why these particular tasks are good ones in order to explore modeling arbitrary-sized observation spaces. [...]  It seems that some domains like sequential decision making in healthcare, or some kind of variable-sized text prediction task would be equally fine [...]
>
> We chose these benchmarks because we were unable to find existing public datasets with irregular observation spaces. Multi-agent modelling problems like those chosen are good examples of spatio-temporal modelling tasks with irregular observations, and also of interest in the machine learning community.
>
> We have been unable to find medical datasets with an irregular observation space like our multi-agent interaction tasks. Most settings here (eg. EEG, Imaging and Tabular data) all exhibit regular shapes (1D, 2D or 3D signal processing). Text based tasks are often treated as a 1D signal, GPT generates token by token until [EOL], BERT reduces a sequence to 1 token, so creating a benchmark here feels somewhat contrived. As mentioned by the reviewer, Shared Autonomy is a domain that is conducive to what we are working on, however common benchmarks from Waymo Open Motion Dataset, ArgoVerse2 and nuScenes are all very short sequences, a few seconds of history to produce a few seconds of prediction. We use our own environments to test modelling over longer sequences, although as mentioned in the limitations, we observe that the amount of context needed saturates around 15 timesteps in (~10sec real time), while motion prediction usually works with 1-5sec history. We have included a mention of motion forecasting datasets in the Discussion and Limitations section.
>
> > Another big issue that I see is that the paper does not make a strong enough case for the choice of baselines. I would expect there to be accompanying work in the domain previously with which to compare against, but because 4.1.2 does not cite any prior work, it's unclear to me that these choices are well-founded. Certainly I'm familiar with Mamba2, but the rest is unclear. Also the choice of encoders for the RNN methods feels somewhat unjustified.
>
>
> We have improved the motivation and clarity around the choice of encoders and dynamics models. Our core focus is on determining appropriate encoding strategies and complementary dynamics models, with efficiency in mind. Baselines and encoding/ dynamics models were chosen primarily for computational complexity reduction and our core goal was to explore the compute performance trade-offs for a range of architectures tackling the problem of spatio-temporal modelling with irregular observations.As such, each of the baselines represent common modelling meta-categories, Recurrent, Transformer and SSM. Given the lack of prior work in the irregular observation handling case, we selected baselines that we felt were appropriate points of comparison. We have added RetNet to represent these linearized transformers (as mentioned in Q6). We have also added citations and additional context for spatio-temporal transformer. Furthermore, we conducted an ablation study on all the spatial encoding methods and found X-Attn Sequential to have the highest accuracy. For consistency, we have now used that algorithm for all the models in the paper.
>
>
> > The last major issues I see is that the results in Figure 5 and Table 1 feel somewhat inconclusive. [...] the fact that it does not appear to have particular advantages in computation that was claimed as a contribution ("efficiently address sequence modelling" at the end of Section 1), makes it hard to understand the contribution.
>
> Our results primarily set out to explore the trade-offs between efficiency and performance in the irregular observation case. This means results should be interpreted given the compute budget available to a practitioner. If only performance matters, practitioners should select a temporal transformer with sequential cross attention (X-Attn Sequential) for observation encoding. For efficiency and model parsimony, an inclusive scan should be selected with X-Attn Sequential. We have made this clearer in the text, and by adding an additional experiment where we directly replace the weighted cumsum with GRU or GILR. This experiment shows a more direct correlation of accuracy with temporal model sophistication and cost. We also changed to a Scan $3\times$ model in SC2 which now has the best training throughput, median memory usage with slight disadvantage in accuracy.
>
> > Section 4.1.2 add a citation to Mamba2
>
> Done
>
> > Section 4.2 it would be nice to have the problem written out mathematically as I find the current description a bit hard to follow
>
> While we moved section 4.2 to the appendix due to space constraints, we have added formulas that represent the decoding algorithms.
>
> TBC.

---

> ### Author Response · Authors · 2024-11-26
>
> Cont.
>
> > Section 5 is missing comment on hyperparameter tuning, especially for baselines
>
> We have added a section in the supplementary on varying the latent state size for chasing-targets with the temporal transformer encoder. While we explored the learning rate on Scan $4\times$ (5e-3, 1e-3, 5e4, 1e-4), we didn’t feel it was particularly pertinent to include this. We did not explore different learning rate schedules.
>
> > Fig 7 should be referenced in 5.1.1
> Done
>
> > Section 3.2 Where exactly does the StarCraft data come from? It would be nice to have a detailed description of the data collection and formatting process in the Appendix
>
> We have added a footnote with a link to the source of the raw data. As mentioned in the reproducibility section, the StarCraft II data collection and conversion tool is made by us and publicly available so is not currently linked for anonymity.
>
> > How different is this method of encoding observations from previous work on StarCraft (regardless of the exact task, the encoding stage should be similar).
>
> The most prominent SC2 model, AlphaStar, used an LSTM for temporal recurrency. AlphaStar-Unplugged uses Transformer XL with up to 2*10^10 frame memory, finding that in pure behaviour cloning LSTM is detrimental compared to no memory. In both models, units are initially encoded with a transformer, then “summarised” into a single feature vector with an averaging operation, and merged with other contextual observation features. This method isn’t particularly sophisticated and we have shown that for a simpler task, chasing-targets, that memory and a larger capacity latent state greatly improves performance.
>
> > What is the "learned initialization" in Section 4.1.2
>
> The initial state of the recurrent neural network, we have adjusted the language to improve clarity.
>
> > Is there some reason why baselines couldn't be matched on some axis? (parameter count is probably easiest)
>
> We matched by the number of layers - 2. We have added this note in the first paragraph of the results section. Models such as the transformer naturally have significantly more parameters to project Q,K,V + Linear FeedForward compared to our method which has zero. This is illustrated in our new 5.2 section that replaces the weighted sum with GILR and GRU, each of which doubles the parameter count (0.9->2->4M), which comes with an associated performance increase. We believe this is a good illustration of scaling between parameters in sequence modelling.
>
> > What I understand in Section 3.2 is that the StarCraft task is a 3-class (idle, move to position, move to enemy) classification problem given sequential context, but still this doesn't exactly specify the setup for me, could you clarify what exactly is the task?
>
> The task is to predict what a player is doing at a given point in time: attacking an enemy unit (assignment to unit), moving to a position (position prediction + position prediction is valid) or idle (no assignment and position prediction is invalid), given the observations of unit positions and their attributes and the context of the minimap. We have revised the wording in Section 3 to improve clarity.
>
> > Section 7 Given the related work (especially Sun et al. 2023) is there some reason why none of these were used as baselines?
>
> We have added RetNet to represent this family of linearized transformers.
>
> > The limitations seem a bit unfortunate, but also foreseeable, is there some reason that the authors specifically designed their own benchmarks instead of working with something longer-term?
>
> We note that our benchmarks are longer duration than many existing (eg. multi-agent tracking). However, as mentioned in the discussion and limitations, models such as these are not really suited to long horizon forecasting. This is a common concern with autoregressive state space models [2]. Our core focus is on architectures that efficiently encode complex irregular observations into a latent and how to pair these with efficient state space models, so it is likely that our approach will inherit these weaknesses.
>
> [2] Appendix B.1 Long-term Forecasting with TiDE: Time-series Dense Encoder https://openreview.net/forum?id=pCbC3aQB5W

---

> > ### Comment · Reviewer_rRfx · 2024-11-30
> > **Thank you for your rebuttal**
> >
> > Thank you for the rebuttal addressing my comments. I will go point by point below.
> >
> > > One large issue I see with the paper is that I'm not totally certain why these particular tasks are good ones in order to explore modeling arbitrary-sized observation spaces. [...] It seems that some domains like sequential decision making in healthcare, or some kind of variable-sized text prediction task would be equally fine [...]
> >
> > Thank you for clarifying. It would be good to provide references to all of these datasets to further bolster your choices, both in the related work (lines 501-515) and in an abbreviated fashion when initially introducing the task (lines 98-132).
> >
> > > Another big issue that I see is that the paper does not make a strong enough case for the choice of baselines. I would expect there to be accompanying work in the domain previously with which to compare against, but because 4.1.2 does not cite any prior work, it's unclear to me that these choices are well-founded. Certainly I'm familiar with Mamba2, but the rest is unclear. Also the choice of encoders for the RNN methods feels somewhat unjustified.
> >
> > I appreciate the improved situation around citation and the addition of the RetNet baseline given how related it is (from the paper's own discussion). Thank you.
> >
> > > The last major issues I see is that the results in Figure 5 and Table 1 feel somewhat inconclusive. [...] the fact that it does not appear to have particular advantages in computation that was claimed as a contribution ("efficiently address sequence modelling" at the end of Section 1), makes it hard to understand the contribution.
> >
> > I still think this is a problem in the updated version. There is still no control along any individual axis, so what I'm left with is a mix of methods. The situation is further confused by discussion around optimized implementations, and how much of a speedup can be reasonably achieved by optimized implementations (like this paper does for Scan) vs. how much of the underlying speed/memory consumption is simply unavoidable. As such, even though parameters are a poor proxy for complexity I think it would be good to control along that axis.
> >
> > > Section 4.1.2 add a citation to Mamba2
> >
> > Thank you.
> >
> > > Section 4.2 it would be nice to have the problem written out mathematically as I find the current description a bit hard to follow
> >
> > Thank you for this change, even though it is now not a central part of the message.
> >
> > > Section 5 is missing comment on hyperparameter tuning, especially for baselines
> >
> > This is important as it's easy to beat weak baselines, and it is easy to get weak baselines by picking a configuration once and leaving it alone forever. I'm still not sure from the response how baselines were tuned.
> >
> > > Fig 7 should be referenced in 5.1.1
> >
> > Thank you.
> >
> > > Section 3.2 Where exactly does the StarCraft data come from? It would be nice to have a detailed description of the data collection and formatting process in the Appendix
> >
> > Thank you, I still think it is useful to have a relatively low-level description in text of how the data is collected, to ground the reader. I personally had difficulty understanding what exactly were the inputs and outputs, and having a description of the data format would help that.
> >
> > > How different is this method of encoding observations from previous work on StarCraft (regardless of the exact task, the encoding stage should be similar).
> >
> > Thank you for this response.
> >
> > > What is the "learned initialization" in Section 4.1.2
> >
> > I think the change is helpful, thank you.
> >
> > > Is there some reason why baselines couldn't be matched on some axis? (parameter count is probably easiest)
> >
> > I'm not sure matching by layers makes sense with such large discrepancies. The reason to do this matching is to make it simple to compare methods. Without any such axis (described above as well), it's hard for me to make any judgment as to which method is pareto-optimal.
> >
> > > What I understand in Section 3.2 is that the StarCraft task is a 3-class (idle, move to position, move to enemy) classification problem given sequential context, but still this doesn't exactly specify the setup for me, could you clarify what exactly is the task?
> >
> > What exactly are the inputs and outputs here? I still don't think the wording is clear in Section 3, though it is approaching clear.
> >
> > > Section 7 Given the related work (especially Sun et al. 2023) is there some reason why none of these were used as baselines?
> >
> > Thank you, I think this is a good choice to include.
> >
> > > The limitations seem a bit unfortunate, but also foreseeable, is there some reason that the authors specifically designed their own benchmarks instead of working with something longer-term?
> >
> > I mostly found the discussion of benchmarks above and why they were unsuitable helpful for answering this question. It would be good to recap some of this discussion in Section 6.

---

> > > ### Author Response · Authors · 2024-12-02
> > >
> > > We thank rRfx for their continued engagement with our paper to improve its quality and the clarity of our contribution.
> > >
> > > > I still think this is a problem in the updated version. There is still no control along any individual axis, so what I'm left with is a mix of methods. The situation is further confused by discussion around optimized implementations, and how much of a speedup can be reasonably achieved by optimized implementations (like this paper does for Scan) vs. how much of the underlying speed/memory consumption is simply unavoidable. As such, even though parameters are a poor proxy for complexity I think it would be good to control along that axis.
> > >
> > > > I'm not sure matching by layers makes sense with such large discrepancies. The reason to do this matching is to make it simple to compare methods. Without any such axis (described above as well), it's hard for me to make any judgment as to which method is pareto-optimal.
> > >
> > > We have added an experiment where we attempt to fix parameter numbers to make this clearer. We note that our baselines are also heavily optimised, in some instances relying on custom cuda implementations (e.g. Mamba-2 or Flash-Attention). As the scan encoder is a parameter free module, we cannot scale the parameter count of the other models down to match Scan 2/4x without reducing the latent state size. However, this has the added benefit of lower memory and training step time as we are not iterating over N temporal encoders for each latent state vector. * indicates the original 2x spatial + 2x temporal encoder pipeline, others are 1x + 1x. In general, Top1 performance follows network capacity (params), but the scan operation appears to provide a moderately better performance capacity tradeoff.
> > > | Encoder | Nx128 Latent | M Param | Train Mem GB | Train Step ms | Top1 |
> > > | - | - | - | - | - | - |
> > > | XS-TT | 1 | 1.107  | 0.49 | 9.0 | 0.602 |
> > > | XS-GRU | 2 | 0.707 | 0.48 | 9.0 | 0.629 |
> > > | Scan 2x | 8 | 0.512 | 0.90 | 14.5 | 0.649 |
> > > | XS-GRU | 4 | 0.906 | 0.63 | 9.6 | 0.663 |
> > > | Scan 4x | 8 | 0.911 | 1.86 | 24.1 | 0.677 |
> > > | XS-GRU | 8 | 1.303 | 0.84 | 10.6 | 0.696 |
> > > | XS-GRU* | 8 | 2.494 | 1.38 | 16.6 | 0.726 |
> > >
> > > > This is important as it's easy to beat weak baselines, and it is easy to get weak baselines by picking a configuration once and leaving it alone forever. I'm still not sure from the response how baselines were tuned.
> > >
> > > We performed the same learning rate sweep with the baseline GRU encoder from Figure 6 and found the same trend as a Scan encoder (see below). We can repeat this for other encoders if requested, pre-empting the reviewer and running a learning rate sweep for model types seemes excessive to us. Collectively, our ablations also cover a broad range of latent sizes so we are confident these are rigorous and strong baselines.
> > >
> > > Avg. Top1 Chasing-Targets
> > > | LR | 5e-3 | 1e-3 | 5e-4 | 1e-4 |
> > > | - | - | - | - | - |
> > > | GRU | 0.513 | 0.726 | 0.718 | 0.656 |
> > > |Scan 4x | 0.642 | 0.688 | 0.675 | 0.576 |
> > >
> > > > What exactly are the inputs and outputs here? I still don't think the wording is clear in Section 3, though it is approaching clear.
> > >
> > > We will add a new section in the appendix “Observation Encoding”, detailing the observation space of the Chasing-Targets and StarCraft II tasks and how they are encoded into feature vectors for the model.
> > > Chasing-Targets uses a sinusoidal position encoding of (x,y,$\theta$) with 16 frequencies per position embedding.
> > >
> > > `agent/particle = cat(enc(x),enc(y),enc(th))`.
> > >
> > > SC2 uses the same position embedding, but also includes a learned embedding of the unit type (dim=29) and the normalized health+shields of the unit.
> > >
> > > `SC2Unit = cat(enc(x),enc(y),enc(th),norm_fn(health,max_health,shield,max_shield),embed[unitTypeId])`.
> > >
> > > In both tasks, we have two groups of agents/particles and SC2Units from the player and enemy. While the agents/particles are fixed over the sequence, the number of SC2Units can vary as they enter and exit the player’s view.
> > > The minimap context is the height-map normalized to [0,1]. Line 212-215 describes how this is encoded and added to the spatio-temporal features.
> > > The output of the model is covered in the “Task Decoding” section of the appendix.
> > >
> > > > Thank you for clarifying. It would be good to provide references to all of these datasets to further bolster your choices, both in the related work (lines 501-515) and in an abbreviated fashion when initially introducing the task (lines 98-132).
> > >
> > > We discussed these datasets in our limitations section in the revised version (Line 456-461). We will add references to these datasets in Section 7 and also find a place to appropriately introduce or mention these earlier in Section 1 or 2.
> > >
> > > > I mostly found the discussion of benchmarks above and why they were unsuitable helpful for answering this question. It would be good to recap some of this discussion in Section 6.
> > >
> > > We added a brief discussion on long horizon forecasting with Transformer/Linear vs SSM/RNN in the revised version (Line 462-465).

---

> > > > ### Comment · Reviewer_rRfx · 2024-12-02
> > > > **Thank you for your response**
> > > >
> > > > > I still think this is a problem in the updated version. There is still no control along any individual axis, so what I'm left with is a mix of methods. The situation is further confused by discussion around optimized implementations, and how much of a speedup can be reasonably achieved by optimized implementations (like this paper does for Scan) vs. how much of the underlying speed/memory consumption is simply unavoidable. As such, even though parameters are a poor proxy for complexity I think it would be good to control along that axis.
> > > >
> > > > Thank you for your answer on this front, and the results controlling along the parameter axis. This I think is a much clearer picture upon which to judge, given all the variability. I now can see that the Scan method is better for parameter usage/performance, but worse along time to compute and training memory. This is helpful.
> > > >
> > > > > This is important as it's easy to beat weak baselines, and it is easy to get weak baselines by picking a configuration once and leaving it alone forever. I'm still not sure from the response how baselines were tuned.
> > > >
> > > > I appreciate the additional tuning which shows the trends are the same. I agree that it seems unlikely that anything drastic will change, but I disagree that it is excessive to tune hyperparameters for all baselines. That seems like basic scientific hygiene.
> > > >
> > > > > What exactly are the inputs and outputs here? I still don't think the wording is clear in Section 3, though it is approaching clear.
> > > >
> > > > Thank you for clarifying this. This is extremely helpful, I hope this will make it into the revision.
> > > >
> > > > > Thank you for clarifying. It would be good to provide references to all of these datasets to further bolster your choices, both in the related work (lines 501-515) and in an abbreviated fashion when initially introducing the task (lines 98-132).
> > > >
> > > > Thank you for the additions, I think this will help the presentation.
> > > >
> > > > > I mostly found the discussion of benchmarks above and why they were unsuitable helpful for answering this question. It would be good to recap some of this discussion in Section 6.
> > > >
> > > > Thank you.
> > > >
> > > > Thanks to the authors for their discussion. I believe I've gained a significantly stronger understanding of the paper as a result. I would vote to accept based on this conversation.

---

> > > > > ### Author Response · Authors · 2024-12-03
> > > > >
> > > > > Thank you for your feedback and updating your score. We have completed the learning rate sweep for the remainder of the models from Figure 6, and found 1e-3 as the best learning rate. The 81 step sequence was accidentally used for GRU, rather than 41, in the previously provided table, this has been rectified.
> > > > >
> > > > > | LR | 5e-3 | 1e-3 | 5e-4 |
> > > > > | - | - | - | - |
> > > > > | GRU | 0.446 | 0.726 | 0.716 |
> > > > > | Scan 4x | 0.642 | 0.688 | 0.675 |
> > > > > | STT | 0.379 | 0.774 | 0.715 |
> > > > > | TT | 0.241 | 0.838 | 0.828 |
> > > > > | Mamba | 0.522 | 0.755 | 0.738 |
> > > > > | RetNet | 0.743 | 0.749 | 0.739 |

---

### Official Review · Reviewer_ttpp · 2024-11-02

**Soundness:** 4
**Presentation:** 2
**Contribution:** 3
**Rating:** 6
**Confidence:** 3

**Summary:**

This paper investigates how to combine the inference efficiency of recurrent models and the training parallelism of multi-head attention transformers, especially for dealing with the varying size observation space. The authors propose a novel algorithm to cycle between cross-attention and inclusive scan to efficiently accumulate historical information. They evaluate their method on two benchmarks, Chasing-Targets gymnasium and StarCraft2.

**Strengths:**

The proposed method, especially integrating inclusive scan algorithm, is novel and is an effective way to accumulate history.

The experiment results are good. The proposed method improves the performance and reduce the computation overhead.

**Weaknesses:**

Please refer to the Questions part.

**Questions:**

I’m not very familiar with this topic, so my questions are mainly about the application settings.

1.	The observations of Chasing-target and StarCraft2 are all low-dimensional. How to extend your method to more realistic and complex scenarios, e.g., vision-based navigation tasks?

2.	How can this method be applied to longer-horizon tasks?

3.	I would like to see more ablation studies on the selection of hyperparameter $\gamma$.

---

> ### Author Response · Authors · 2024-11-26
>
> > observations of Chasing-target and StarCraft2 are all low-dimensional
>
> There is already substantial work on tasks with fixed-dimensional observations like vision-based navigation tasks. Our work focuses on a less commonly studied setting of variable dimension observations. These are often low-dimensional, but with complex data structures that are often seen in real-world settings. For example, in our starcraft data, the observations not only include variable numbers of units and positions, but also their health and the unit type (as a learned embedding). This problem setting is very realistic, for example in inventory management, digital twin simulations or tabular data structures with missing data. That being said, high dimensional observations could be trivially included in the proposed architecture, by tokenising the scene (eg. with image chips or object tokens from a DETR-like vision encoder).
>
> > How can this method be applied to longer-horizon tasks?
>
> Although our benchmarks do not explicitly test longer horizon modelling, the method can still be applied to these tasks directly. There is no particular limitation on the input sequence length, like any other recurrent model. Training requires O(L) memory whereas inference is O(1). However, as mentioned in our response to Reviewer aVTi, autoregressive models are known to perform worse in longer horizon prediction tasks, so our approach is likely to inherit these weaknesses.
>
> > I would like to see more ablation studies on the selection of hyperparameter (Gamma)
>
> We have added gamma=1.5 and 3 and created a separate figure in the supplementary material plotting all the curves (1,1.5,2,3). gamma=2 retains the highest performance, followed by 1.5, 3 and 1. We have also added baselines to GILR, a state-space model that can be formulated as an inclusive scan with an input-conditioned gamma parameter.

---

### Official Review · Reviewer_aVTi · 2024-11-12

**Soundness:** 2
**Presentation:** 2
**Contribution:** 2
**Rating:** 6
**Confidence:** 4

**Summary:**

The paper discusses a approach to spatio-temporal modelling for tasks with irregular observation spaces. Experiments are conducted on multi-agent scenario's like robotic target-chasing simulations and StarCraft II gameplay analysis.


Motivation: The paper aruges that current architectures like Transformers and LSTMs works well in scenarios with fixed observation spaces but struggle with tasks where observation spaces vary over time, such as multi-agent interactions. The paper aims to combine the efficiency of recurrent models with the parallelism of attention mechanisms, focusing on varying-dimension observation spaces.

Proposed Method:  The paper introduce an algorithm using a 2D latent state, alternating between cross-attention on observations and a discounted cumulative sum over time, capturing long context.

Experiments: Two benchmarks are used for evaluation: (a) Chasing Targets: Agents pursuing dynamic targets, evaluating the model's ability to infer intended goals in real-time; (b) StarCraft II: A environment with complex, temporally varying observations, where models infer combat unit assignments and predict unit movement.

Results: The proposed model achieves competitive accuracy with fewer parameters and faster inference compared to standard methods (e.g., transformers). The "resampling cycle," improves sequence modeling accuracy by conditioning observations on accumulated sequence data.

===================

After rebuttal: I've read the review by other reviewers as well as the rebuttal by the authors.

This statement by the authors summarizes the contribution of the paper well.
"Our results primarily set out to explore the trade-offs between efficiency and performance in the irregular observation case. This means results should be interpreted given the compute budget available to a practitioner. If only performance matters, practitioners should select a temporal transformer with sequential cross attention (X-Attn Sequential) for observation encoding. For efficiency and model parsimony, an inclusive scan should be selected with X-Attn Sequential."

It will be very helpful to revise the introduction and abstract accordingly.

**Strengths:**

The paper investigates the use of various encoders to encode irregular observations into a fixed dimensional latent, and corresponding sequence modelling approaches.

**Weaknesses:**

- The model's efficiency enables its use in real-time applications with irregular observation space, but studying the effect on longer sequence durations as well as diversifying the tasks (i.e., conduct experiments on more tasks) would further validate its utility in long-term planning tasks.
- There's some work in the literature like Temporal Latent Bottleneck (https://arxiv.org/abs/2205.14794) and Block Recurrent Transformers (https://arxiv.org/abs/2203.07852); which combines strengths of attention and recurrence.

**Questions:**

See Weaknesses.

---

> ### Author Response · Authors · 2024-11-26
>
> > ...studying the effect on longer sequence durations as well as diversifying the tasks (i.e., conduct experiments on more tasks) would further validate its utility in long-term planning tasks.
>
> We note that our benchmarks are longer duration than many existing (eg. Waymo Open Motion Dataset). However, as mentioned in the discussion and limitations, models such as these are not really suited to long horizon forecasting. This is a common concern with autoregressive state space models [2]. Our core focus is on architectures that efficiently encode complex irregular observations into a latent and how to pair these with efficient state space models, so it is likely that our approach will inherit these weaknesses.
>
> [2] Appendix B.1 Long-term Forecasting with TiDE: Time-series Dense Encoder https://openreview.net/forum?id=pCbC3aQB5W
>
>
> > There's some work in the literature like Temporal Latent Bottleneck [...]  and Block Recurrent Transformer…
>
> We thank the reviewer for pointing us to another strategy for sub-quadratic transformer modelling, we have included discussion of this in our Related Work section. While we have not included the mentioned block+recurrency methods in our comparison studies, we have added RetNet to represent linear attention models of this form.

---

### Author Response · Authors · 2024-11-26

We thank the reviewers for their targeted and thoughtful comments to improve our paper. We have added additional experiments and sections to the paper to include the additional requests such as including RetNet, more gamma ablations, and consistency with spatial encoder choice. Furthermore, we have added a direct comparison between our weighted summation and two gated recurrent unit variants, facilitated by directly swapping scan with GRU or GILR [1]. With these additional experiments, we have strengthened our introduction and contributions. To fit this within the page limit, we have moved the task decoding to the appendix as while it is still important for the understanding of the model, it is not pertinent to the core results of the paper.

[1] Parallelizing Linear Recurrent Neural Nets Over Sequence Length https://openreview.net/forum?id=HyUNwulC-

---

> ### Author Response · Authors · 2024-11-26
> **New Data**
>
> We recommend reviewing the plots within the revised manuscript which have been updated, but here are tables with new ablations mentioned in the comment above. Other new ablations such as latent state size are plots in the appendix of the revised paper.
>
> ## Spatial-Encoder with Temporal-Only Transformer
>
> |    Encoder      |   Algorithm           | Top1 Acc.  |  # M params| Mem. GB | Train it/sec| Infer. it/sec |
> |---------|------------|-------|-------|------|-------|--------|
> | BERT    | piecewise  | 0.769 | 10.4  | 2.88 | 38.8  | 62.7   |
> | BERT    | fused      | 0.789 | **9.99** | 2.98 | 44.4  | 66.5   |
> | BERT    | sequential | 0.809 | 10.4  | 3.64 | 35.5  | 59.3   |
> | X-Attn  | piecewise  | 0.805 | 10.4  | 2.44 | 41.7  | 58.5   |
> | X-Attn  | fused      | 0.820 | **9.99** | **2.19** | **46.2** | **71.3** |
> | X-Attn  | sequential | **0.838** | 10.4  | 2.90 | 39.4  | 53.3   |
>
>
> ## Replacing Scan with Gating (4-Block)
>
> | Encoder | Top1 Acc. | # M params | Mem. GB | Train it/sec | Infer. it/sec |
> |---------|-----------|-------------|---------|--------------|---------------|
> | Scan    | 0.680     | **0.912**   | **3.16**| **22.6**     | **23.3**      |
> | GILR    | 0.714     | 1.97        | 4.490   | 13.9         | 17.2          |
> | GRU     | **0.734** | 4.082       | 3.85    | 20.2         | 20.9          |
>
> ## Updated SC2 Evalauation
>
> | Encoder     | Top1 Acc. | Top1 +null | Top5 Acc. | MSE  | F1   | # M Params | Mem. GB | Train it/sec |
> |-------------|-----------|------------|-----------|------|------|------------|---------|--------------|
> | LSTM        | 0.216     | 0.905      | 0.637     | 0.190| 0.811| 32.1       | **9.59**| 14.6         |
> | TT          | **0.258** | **0.918**  | **0.647** | **0.169**| **0.844**| 57.4 | 10.6    | 14.7         |
> | RetNet      | 0.156     | 0.916      | 0.622     | 0.178| 0.834| 32.1       | 10.0    | 9.8          |
> | Mamba2      | 0.238     | 0.913      | 0.643     | 0.180| 0.832| 29.1       | **9.59**| 14.3         |
> | Scan $3\times$ | 0.209  | 0.892      | 0.632     | 0.202| 0.796| **14.5**   | 9.91    | **15.1**     |
>
> ## Recurrent Chasing-Targets Assignment
> | Encoder | Init. | Top1 Acc. | # M Param. | Mem. GB | Train it/sec | Infer it/sec |
> |---------|-------|-----------|------------|---------|--------------|--------------|
> | RNN     | Zero  | 0.569     | **1.44**   | **1.31**| 62.2         | **69.1**     |
> | RNN     | Learn | 0.571     | **1.44**   | 1.33    | **62.1**     | **69.1**     |
> | GRU     | Zero  | 0.720     | 2.49       | 1.54    | 59.5         | 64.9         |
> | GRU     | Learn | **0.726** | 2.49       | 1.38    | 60.6         | 65.7         |
> | LSTM    | Zero  | 0.715     | 3.02       | 1.46    | 55.5         | 62.1         |
> | LSTM    | Learn | 0.707     | 3.02       | 1.49    | 55.5         | 61.4         |

---

### Author Response · Authors · 2024-12-03

We would once again like to thank the reviewers for their valuable feedback which has helped us improve our paper. A summary of the key changes are as follows:

- Added evaluation of spatial encoding combinations, standardized usage of X-Attn Sequential
- Added learning rate sweep for chasing-targets temporal encoders
- Added gamma sweep for chasing-targets Scan 4x
- Included a new experiment, swapping the Scan module with GILR or GRU.
- Added RetNet and Transformer (per latent, “TT”) baselines to experiments
- Added discussion on motion forecasting datasets
- Added discussion on long-term forecasting challenges
- Improved clarity in various areas pointed out by reviewers

We believe that our contributions, introducing a framework for efficiently handling spatio-temporal challenges with irregular observations, a simple, yet effective method for accumulating temporal context on the back of extensive ablations and evaluations of efficient architectures alongside two challenging tasks to demonstrate model capability, are valuable to the community.

---

### Meta-Review · Area_Chair_rBXh · 2024-12-23

**Metareview:**

The paper presents an interesting approach to sequence modeling with irregular observation spaces, exploring novel scan operations and cross-attention techniques for handling variable-dimensional observations. While the work demonstrates potential in multi-agent tasks and introduces innovative modeling strategies, it struggles with comprehensive validation and generalizability. Reviewers noted significant challenges in methodology, including unclear motivation and insufficient differentiation from existing approaches. The benchmarks, primarily based on custom multi-agent tasks, do not definitively prove the method's broader applicability, and performance improvements remain marginal. Despite the authors' efforts to address concerns through additional experiments and clarifications, fundamental limitations persist. The incremental nature of the contribution, combined with presentation challenges and limited experimental rigor, suggests that while the work shows promise, it is not yet sufficiently developed for publication at a top-tier conference.

**Additional Comments On Reviewer Discussion:**

Reviewers raised concerns about the paper's methodology while acknowledging its innovative approach to handling irregular observation spaces. The authors demonstrated potential through novel scan operations and cross-attention techniques, responding to critiques by adding comprehensive ablation studies, more detailed citations, and clarifying experimental motivations. Despite introducing interesting multi-agent benchmarks and showing promise in computational trade-offs, reviewers remained uncertain about the method's broader applicability. The authors expanded their experimental validation, controlled parameter studies, and provided more nuanced explanations of their approach.

---

### Decision · Program_Chairs · 2025-01-22

Reject